# AdvFlow: Inconspicuous Black-box Adversarial Attacks using Normalizing Flows

**Hadi M. Dolatabadi, Sarah Erfani, Christopher Leckie**
School of Computing and Information Systems
The University of Melbourne
Parkville, Victoria, Australia
hadi.mohagheghdolatabadi@student.unimelb.edu.au

## Abstract

Deep learning classifiers are susceptible to well-crafted, imperceptible variations of their inputs, known as adversarial attacks. In this regard, the study of powerful attack models sheds light on the sources of vulnerability in these classifiers, hopefully leading to more robust ones. In this paper, we introduce AdvFlow: a novel black-box adversarial attack method on image classifiers that exploits the power of normalizing flows to model the density of adversarial examples around a given target image. We see that the proposed method generates adversaries that closely follow the clean data distribution, a property which makes their detection less likely. Also, our experimental results show competitive performance of the proposed approach with some of the existing attack methods on defended classifiers. The code is available at https://github.com/hmdolatabadi/AdvFlow.

## 1 Introduction

Deep neural networks (DNN) have been successfully applied to a wide variety of machine learning tasks. For instance, trained neural networks can reach human-level accuracy in image classification [46]. However, Szegedy et al. [53] showed that such classifiers can be fooled by adding an imperceptible perturbation to the input image. Since then, there has been extensive research in this area known as *adversarial machine learning*, trying to design more powerful attacks and devising more robust neural networks. Today, this area encompasses a broader type of data than images, with video [25], graphs [65], text [34], and other types of data classifiers being attacked.

In this regard, the design of stronger adversarial attacks plays a crucial role in understanding the nature of possible real-world threats. The ultimate goal of such studies is to help neural networks become more robust against such adversaries. This line of research is extremely important as even the slightest flaw in some real-world applications of DNNs such as self-driving cars can have severe, irreparable consequences [12].

In general, adversarial attack approaches can be classified into two broad categories: white-box and black-box. In *white-box* adversarial attacks, the assumption is that the threat model has full access to the target DNN. This way, adversaries can leverage their knowledge about the target model to generate adversarial examples (for instance, by taking the gradient of the neural network). In contrast, *black-box* attacks assume that they do not know the internal structure of the target model a priori. Instead, they can only *query* the model about some inputs, and work with the labels or confidence levels associated with them [62]. Thus, black-box attacks seem to be making more realistic assumptions. In the beginning, black-box attacks were mostly thought of as the transferability of white-box adversarial examples to unseen models [42]. Recently, however, there has been more research to attack black-box models directly.

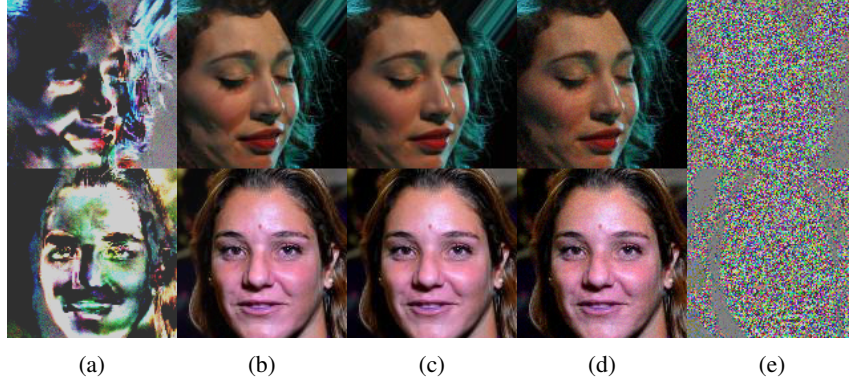

|     |     |     |     |     |
| --- | --- | --- | --- | --- |
| (a) | (b) | (c) | (d) | (e) |

Figure 1: Adversarial perturbations generated by AdvFlow take the structure of the original image into account, resulting in less detectable adversaries compared to $\mathcal{N}$ATTACK [33] (see Section 4.1). The classifier is a VGG19 [50] trained to detect smiles in CelebA [36] faces. (a) AdvFlow magnified difference (b) AdvFlow adversarial example (c) clean image (d) $\mathcal{N}$ATTACK adversarial example (e) $\mathcal{N}$ATTACK magnified difference.

In this paper, we introduce AdvFlow: a black-box adversarial attack that makes use of pre-trained normalizing flows to generate adversarial examples. In particular, we utilize flow-based methods pre-trained on clean data to model the probability distribution of possible adversarial examples around a given image. Then, by exploiting the notion of *search gradients* from *natural evolution strategies (NES)* [59, 58], we solve the black-box optimization problem associated with adversarial example generation to adjust this distribution. At the end of this process, we wind up having a data distribution whose realizations are likely to be adversarial. Since this density is constructed on the top of the original data distribution estimated by normalizing flows, we see that the generated perturbations take on the structure of data rather than an additive noise (see Figure 1). This property impedes distinguishing AdvFlow examples from clean data for adversarial example detectors, as they often assume that the adversaries come from a different distribution than the clean data. Moreover, we prove a lemma to conclude that adversarial perturbations generated by the proposed approach can be approximated by a normal distribution with dependent components. We then put our model under test and show its effectiveness in generating adversarial examples with 1) less detectability, 2) higher success rate, 3) lower number of queries, and 4) higher rate of transferability on defended models compared to the similar method of $\mathcal{N}$ATTACK [33].

In summary, we make the following contributions:

- We introduce AdvFlow, a black-box adversarial attack that leverages the power of normalizing flows in modeling data distributions. To the best of our knowledge, this is the first work that explores the use of flow-based models in the design of adversarial attacks.

- We prove a lemma about the adversarial perturbations generated by AdvFlow. As a result of this lemma, we deduce that AdvFlows can generate perturbations with dependent elements, while this is not the case for $\mathcal{N}$ATTACK [33].

- We show the power of the proposed approach in generating adversarial examples that have a similar distribution to the data. As a result, our method is able to mislead adversarial example detectors for they often assume adversaries come from a different distribution than the clean data. We then see the performance of the proposed approach in attacking some of the most recent adversarial training defense techniques.

## 2 Related Work

In this section, we review some of the most closely related work to our proposed approach. For a complete review of (black-box) adversarial attacks, we refer the interested reader to [62, 3].

**Black-box Adversarial Attacks.** In one of the earliest black-box approaches, Chen et al. [5] used the idea of Zeroth Order Optimization and came up with a method called *ZOO*. In particular, ZOO

uses the target neural network queries to build up a zero-order gradient estimator. Then, it utilizes the estimated gradient to minimize a Carlini and Wagner (C&W) loss [4] and find an adversarial image. Later and inspired by [58, 47], Ilyas et al. [22] tried to estimate the DNN gradient using a normally distributed search density. In particular, they estimate the gradient of the classifier $\mathcal{C}(\mathbf{x})$ with

$$\nabla_{\mathbf{x}}\mathcal{C}(\mathbf{x}) \approx \mathbb{E}_{\mathcal{N}(\mathbf{z}|\mathbf{x},\sigma^2 I)}\left[\mathcal{C}(\mathbf{z})\nabla_{\mathbf{x}}\log\left(\mathcal{N}(\mathbf{z}|\mathbf{x},\sigma^2 I)\right)\right],$$

which only requires querying the black-box model $\mathcal{C}(\mathbf{x})$. Having the DNN gradient estimate, Ilyas et al. [22] then take a *projected gradient descent (PGD)* step to minimize their objective for generating an adversarial example. This idea is further developed in the construction of $\mathcal{N}$ATTACK [33]. Specifically, instead of trying to minimize the adversarial example generation objective directly, they aim to fit a distribution around the clean data so that its realizations are likely to be adversarial (see Section 3.3 for more details). In another piece of work, Ilyas et al. [23] observe that the gradients used in adversarial example generation by PGD exhibit a high correlation both in time and across data. Thus, the number of queries to attack a black-box model can be reduced if one incorporates this prior knowledge about the gradients. To this end, Ilyas et al. [23] uses a bandit-optimization technique to integrate these priors into their attack, resulting in a method called *Bandits & Priors*. Finally, *Simple Black-box Attack* (SimBA) [16] is a straightforward, intuitive approach to construct black-box adversarial examples. It is first argued that for any particular direction $\mathbf{q}$ and step size $\epsilon > 0$, either $\mathbf{x} - \epsilon\mathbf{q}$ or $\mathbf{x} + \epsilon\mathbf{q}$ is going to decrease the probability of detecting the correct class label of the input image $\mathbf{x}$. Thus, we are likely to find an adversary by iteratively taking such steps. The vectors $\mathbf{q}$ are selected from a set of orthonormal candidate vectors $Q$. Guo et al. [16] use Discrete Cosine Transform (DCT) to construct such a set, exploiting the observation that "random noise in low-frequency space is more likely to be adversarial" [15].

**Adversarial Attacks using Generative Models.**  There has been some prior work that utilizes the power of generative models (mostly generative adversarial networks (GAN)) to model adversarial perturbations and attack DNNs [2, 61, 56, 20]. The target of these models is mainly white-box attacks. They require training of their parameters to produce adversarial perturbations using a cost function that involves taking the gradient of a target network. To adapt themselves to black-box settings, they try to replace this target network with either a distilled version of it [61], or a substitute source model [20]. However, as we will see in Section 3, the flow-based part of our model is only pre-trained on some clean training data using the maximum likelihood objective of Eq. (5). Thus, AdvFlow can be adapted to any target classifier of the same dataset, without the need to train it again. Moreover, while prior work is mainly concerned with generating the adversarial perturbations (for example [55]), here we use the normalizing flows output as the adversarial example directly. In this sense, our work is more similar to [51] that generates unrestricted adversarial examples using GANs in a white-box setting, and falls under functional adversarial attacks [31]. However, besides being black-box, in AdvFlow we restrict the output to be in the vicinity of the original image.

## 3 Proposed Method

In this section, we propose our attack method. First, we define the problem of black-box adversarial attacks formally. Next, we go over normalizing flows and see how we can train a flow-based model. Then, we review the idea of Natural Evolution Strategies (NES) [59, 58] and $\mathcal{N}$ATTACK [33]. Afterward, we show how normalizing flows can be mixed with NES in the context of black-box adversarial attacks, resulting in a method we call AdvFlow. Finally, we prove a lemma about the nature of the perturbations generated by the proposed approach and show that $\mathcal{N}$ATTACK cannot produce the adversarial perturbations generated by AdvFlow. Our results in Section 4 support this lemma. There, we see that AdvFlow can generate adversarial examples that are less detectable than the ones generated by $\mathcal{N}$ATTACK [33] due to its perturbation structure.

### 3.1  Problem Statement

Let $\mathcal{C}(\cdot) : \mathcal{X}^d \to \mathcal{P}^k$ denote a DNN classifier. Assume that the classifier takes a $d$-dimensional input $\mathbf{x} \in \mathcal{X}^d$, and outputs a vector $\mathbf{p} \in \mathcal{P}^k$. Each element of the vector $\mathbf{p}$ indicates the probability of the input belonging to one of the $k$ classes that the classifier is trying to distinguish. Furthermore, let $y$ denote the correct class label of the data. In other words, if the $y$-th element of the classifier output $\mathbf{p}$ is larger than the rest, then the input has been correctly classified. Finally, let the well-known Carlini

and Wagner (C&W) loss [4] be defined as[1]

$$\mathcal{L}(\mathbf{x}') = \max\left(0, \log \mathcal{C}(\mathbf{x}')_y - \max_{c \neq y} \log \mathcal{C}(\mathbf{x}')_c\right), \qquad (1)$$

where $\mathcal{C}(\mathbf{x}')_y$ indicates the $y$-th element of the classifier output. In the C&W objective, we always have $\mathcal{L}(\mathbf{x}') \geq 0$. The minimum occurs when $\mathcal{C}(\mathbf{x}')_y \leq \max_{c \neq y} \mathcal{C}(\mathbf{x}')_c$, which is an indication that our classifier has been fooled. Thus, finding an adversarial example for the input data $\mathbf{x}$ can be written as [33]:

$$\mathbf{x}_{adv} = \arg\min_{\mathbf{x}' \in \mathcal{S}(\mathbf{x})} \mathcal{L}(\mathbf{x}'). \qquad (2)$$

Here, $\mathcal{S}(\mathbf{x})$ denotes a set that contains similar data to $\mathbf{x}$ in an appropriate manner. For example, it is common to define

$$\mathcal{S}(\mathbf{x}) = \left\{\mathbf{x}' \in \mathcal{X}^d \mid \|\mathbf{x}' - \mathbf{x}\|_p \leq \epsilon_{\max}\right\} \qquad (3)$$

for image data. In this paper, we define $\mathcal{S}(\mathbf{x})$ as in Eq. (3) since we deal with the application of our attack on images.

## 3.2 Flow-based Modeling

**Normalizing Flows.** Normalizing flows (NF) [54, 7, 44] are a family of generative models that aim at modeling the probability distribution of a given dataset. To this end, they make use of the well-known *change of variables* formula. In particular, let $\mathbf{Z} \in \mathbb{R}^d$ denote a random vector with a straightforward, known distribution such as uniform or standard normal. The *change of variables* formula states that if we apply an invertible and differentiable transformation $\mathbf{f}(\cdot) : \mathbb{R}^d \to \mathbb{R}^d$ on $\mathbf{Z}$ to obtain a new random vector $\mathbf{X} \in \mathbb{R}^d$, the relationship between their corresponding distributions can be written as:

$$p(\mathbf{x}) = p(\mathbf{z}) \left| \det\left(\frac{\partial \mathbf{f}}{\partial \mathbf{z}}\right) \right|^{-1}. \qquad (4)$$

Here, $p(\mathbf{x})$ and $p(\mathbf{z})$ denote the probability distributions of $\mathbf{X}$ and $\mathbf{Z}$, respectively. Moreover, the multiplicative term on the right-hand side is called the absolute value of the Jacobian determinant. This term accounts for the changes in the volume of $\mathbf{Z}$ due to applying the transformation $\mathbf{f}(\cdot)$. Flow-based methods model the transformation $\mathbf{f}(\cdot)$ using stacked layers of invertible neural networks (INN). They then apply this transformation on a *base random vector* $\mathbf{Z}$ to model the data density. In this paper, we assume that the base random vector has a standard normal distribution.

**Maximum Likelihood Estimation.** To fit the parameters of INNs to the i.i.d. data observations $\mathbf{x}_1, \mathbf{x}_2, \ldots, \mathbf{x}_n$, NFs use the following maximum likelihood objective [44]:

$$\boldsymbol{\theta}^* = \arg\max_{\boldsymbol{\theta}} \frac{1}{n} \sum_{i=1}^{n} \log p_{\boldsymbol{\theta}}(\mathbf{x}_i). \qquad (5)$$

Here, $\boldsymbol{\theta}$ denotes the parameter set of the model and $p_{\boldsymbol{\theta}}$ is the density defined in Eq. (4). Note that INNs should be modeled such that they allow for efficient computation of their Jacobian determinant. Otherwise, this issue can impose a severe hindrance in the application of NFs to high-dimensional data given the cubic complexity of determinant computation. For a more detailed review of normalizing flows, we refer the interested reader to [41, 28] and the references within.

**Training the Flow-based Models.** We assume that we have access to some training data of the same domain to pre-train our flow-based model. However, at the time of adversarial example generation, we use unseen test data. Note that while in our experiments we use the same training data as the classifier itself, we are not obliged to do so. We observed that our results remain almost the same even if we separate the flow-based model training data from what the classifier is trained on. We argue that using the same training data is valid since our flow-based models do not extract discriminative features, and they are only trained on clean data. This is in contrast to other generative approaches used for adversarial example generation [2, 61, 56, 20]. In Appendix C.5, we empirically show that not only is this statement accurate, but we can get almost the same performance by using similar datasets to the true one.

### 3.3 Natural Evolution Strategies and $\mathcal{N}$ATTACK

**Natural Evolution Strategies (NES).** Our goal is to solve the optimization problem of Eq. (2) in a black-box setting, meaning that we only have access to the inputs and outputs of the classifier $\mathcal{C}(\cdot)$. Natural Evolution Strategies (NES) use the idea of *search gradients* to optimize Eq. (2) [59, 58]. To this end, a so-called *search distribution* is first defined, and then the expected value of the original objective is optimized under this distribution.

In particular, let $p(\mathbf{x}'|\boldsymbol{\psi})$ denote the search distribution with parameters $\boldsymbol{\psi}$. Then, in NES we aim to minimize

$$J(\boldsymbol{\psi}) = \mathbb{E}_{p(\mathbf{x}'|\boldsymbol{\psi})}\left[\mathcal{L}(\mathbf{x}')\right] \tag{6}$$

over $\boldsymbol{\psi}$ as a surrogate for $\mathcal{L}(\mathbf{x}')$ [58]. To minimize Eq. (6) using gradient descent, one needs to compute the Jacobian of $J(\boldsymbol{\psi})$ with respect to $\boldsymbol{\psi}$. To this end, NES makes use of the "log-likelihood trick" [58] (see Appendix A.1)

$$\nabla_{\boldsymbol{\psi}} J(\boldsymbol{\psi}) = \mathbb{E}_{p(\mathbf{x}'|\boldsymbol{\psi})}\left[\mathcal{L}(\mathbf{x}')\nabla_{\boldsymbol{\psi}}\log\left(p(\mathbf{x}'|\boldsymbol{\psi})\right)\right]. \tag{7}$$

Finally, the parameters of the model are updated using a gradient descent step with learning rate $\alpha$:[2]

$$\boldsymbol{\psi} \leftarrow \boldsymbol{\psi} - \alpha\nabla_{\boldsymbol{\psi}} J(\boldsymbol{\psi}). \tag{8}$$

$\mathcal{N}$**ATTACK.** To find an adversarial example for an input $\mathbf{x}$, $\mathcal{N}$ATTACK [33] tries to find a distribution $p(\mathbf{x}'|\boldsymbol{\psi})$ over the set of legitimate adversaries $\mathcal{S}(\mathbf{x})$ in Eq. (3). Therefore, it models $\mathbf{x}'$ as

$$\mathbf{x}' = \text{proj}_{\mathcal{S}}\left(\tfrac{1}{2}(\tanh(\mathbf{z}) + 1)\right), \tag{9}$$

where $\mathbf{z} \sim \mathcal{N}(\mathbf{z}|\boldsymbol{\mu}, \sigma^2 I)$ is an isometric normal distribution with mean $\boldsymbol{\mu}$ and standard deviation $\sigma$. Moreover, $\text{proj}_{\mathcal{S}}(\cdot)$ projects its input back into the set of legitimate adversaries $\mathcal{S}(\mathbf{x})$. Li et al. [33] define their model parameters as $\boldsymbol{\psi} = \{\boldsymbol{\mu}, \sigma\}$. Then, they find $\sigma$ using grid-search and $\boldsymbol{\mu}$ by the update rule of Eq. (8) exploiting NES.

### 3.4 Our Approach: AdvFlow

Recently, there has been some effort to detect adversarial examples from clean data. The primary assumption of these methods is often that the adversaries come from a different distribution than the data itself; for instance, see [37, 32, 64]. Thus, to come up with more powerful adversarial attacks, it seems reasonable to construct adversaries that have a similar distribution to the clean data. To this end, we propose *AdvFlow*: a black-box adversarial attack that seeks to build inconspicuous adversaries by leveraging the power of normalizing flows (NF) in exact likelihood modeling of the data [8].

Let $\mathbf{f}(\cdot)$ denote a pre-trained, invertible and differentiable NF model on the clean training data. To reach our goal of decreasing the attack's detectability, we propose using this pre-trained, fixed NF transformation to model the adversaries. In an analogy with Eq. (9), we assume that our adversarial example comes from a distribution that is modeled by

$$\mathbf{x}' = \text{proj}_{\mathcal{S}}\left(\mathbf{f}(\mathbf{z})\right), \qquad \mathbf{z} \sim \mathcal{N}(\mathbf{z}|\boldsymbol{\mu}, \sigma^2 I) \tag{10}$$

where $\text{proj}_{\mathcal{S}}(\cdot)$ is a projection rule that keeps the generated examples in the set of legitimate adversaries $\mathcal{S}(\mathbf{x})$. By the change of variables formula from Eq. (4), we know that $\mathbf{f}(\mathbf{z})$ in Eq. (10) is distributed similar to the clean data distribution. The only difference is that the base density is transformed by an affine mapping, i.e., from $\mathcal{N}(\mathbf{z}|\mathbf{0}, I)$ to $\mathcal{N}(\mathbf{z}|\boldsymbol{\mu}, \sigma^2 I)$. This small adjustment can result in an overall distribution for which the generated samples are likely to be adversarial.

Putting the rule of the *lazy statistician* [57] together with our attack definition in Eq. (10), we can write down the objective function of Eq. (6) as

$$J(\boldsymbol{\mu}, \sigma) = \mathbb{E}_{p(\mathbf{x}'|\boldsymbol{\mu}, \sigma)}\left[\mathcal{L}(\mathbf{x}')\right] = \mathbb{E}_{\mathcal{N}(\mathbf{z}|\boldsymbol{\mu}, \sigma^2 I)}\left[\mathcal{L}\left(\text{proj}_{\mathcal{S}}\left(\mathbf{f}(\mathbf{z})\right)\right)\right]. \tag{11}$$

As in $\mathcal{N}$ATTACK [33], we will consider $\sigma$ to be a hyperparameter.[3] Thus, we are only required to minimize $J(\boldsymbol{\mu}, \sigma)$ with respect to $\boldsymbol{\mu}$. Using the "log-likelihood trick" of Eq. (7), we can derive the Jacobian of $J(\boldsymbol{\mu}, \sigma)$ as

$$\nabla_{\boldsymbol{\mu}} J(\boldsymbol{\mu}, \sigma) = \mathbb{E}_{\mathcal{N}(\mathbf{z}|\boldsymbol{\mu}, \sigma^2 I)} \left[ \mathcal{L}\Big( \mathrm{proj}_{\mathcal{S}}\big(\mathbf{f}(\mathbf{z})\big) \Big) \nabla_{\boldsymbol{\mu}} \log \mathcal{N}(\mathbf{z}|\boldsymbol{\mu}, \sigma^2 I) \right].$$

This expectation can then be estimated by sampling from a distribution $\mathcal{N}(\mathbf{z}|\boldsymbol{\mu}, \sigma^2 I)$ and forming their sample average. Next, we update the parameter $\boldsymbol{\mu}$ by performing a gradient descent step

$$\boldsymbol{\mu} \leftarrow \boldsymbol{\mu} - \alpha \nabla_{\boldsymbol{\mu}} J(\boldsymbol{\mu}, \sigma).$$

In the end, we generate our adversarial example by sampling from Eq. (10).

**Practical Considerations.** To help our model to start its search from an appropriate point, we first transform the clean data to its latent space representation. Then, we aim to find a small additive latent space perturbation in the form of a normal distribution. Moreover, as suggested in [33], instead of working with $\mathcal{L}\big(\mathrm{proj}_{\mathcal{S}}\big(\mathbf{f}(\mathbf{z})\big)\big)$ directly, we normalize them so that they have zero mean and unit variance to help AdvFlows convergence faster. Finally, among different flow-based models, it is preferable to choose those that have a straightforward inverse, such as [8, 27, 11, 10]. This way, we can efficiently go back and forth between the original data and their base distribution representation. Algorithm 1 in Appendix D.1 summarizes our black-box attack method. Other variations of AdvFlow can also be found in Appendix D. These variations include our solution to high-resolution images and investigation of un-trained AdvFlow.

### 3.4.1 AdvFlow Interpretation

We can interpret AdvFlow from two different perspectives.

First, there exists a probabilistic view: we use the flow-based model transformation of the original data, and then try to adjust it using an affine transformation on its base distribution. The amount of this change is determined by our urge to minimize the C&W cost of Eq. (1) such that we get the minimum value on average. Thus, if it is successful, we will end up having a distribution whose samples are likely to be adversarial. Meanwhile, since this distribution is initialized with that of clean data, it resembles the clean data density closely.

Second, we can think of AdvFlow as a search over the latent space of flow-based models. We map the clean image to the latent space and then try to search in the vicinity of that point to find an adversarial example. This search is ruled by the objective of Eq. (11). Since our approach exploits a fully invertible, pre-trained flow-based model, we would expect to get an adversarial example that resembles the original image in the structure and look less noisy. This adjustment gives our model the flexibility to produce perturbations that take the structure of clean data into account (see Figure 1).

### 3.4.2 Uniqueness of AdvFlow Perturbations

In this section, we present a lemma about the nature of perturbations generated by AdvFlow and $\mathcal{N}$ATTACK [33]. As a direct result of this lemma, we can easily deduce that the adversaries generated by AdvFlow can be approximated by a normal distribution whose components are dependent. However, this is not the case for $\mathcal{N}$ATTACK as they always have independent elements. In this sense, we can then rigorously conclude that the AdvFlow perturbations are unique and cannot be generated by $\mathcal{N}$ATTACK. Thus, we cannot expect $\mathcal{N}$ATTACK to be able to generate perturbations that look like the original data. This result can also be generalized to many other attack methods as they often use an additive, independent perturbation. Proofs can be found in Appendix A.2.

**Lemma 3.1.** *Let* $\mathbf{f}(\mathbf{x})$ *be an invertible, differentiable function. For a small perturbation* $\boldsymbol{\delta}_z$ *we have*

$$\boldsymbol{\delta} = \mathbf{f}\big(\mathbf{f}^{-1}(\mathbf{x}) + \boldsymbol{\delta}_z\big) - \mathbf{x} \approx \big(\nabla \mathbf{f}^{-1}(\mathbf{x})\big)^{-1} \boldsymbol{\delta}_z.$$

**Corollary 3.1.1.** *The adversarial perturbations generated by AdvFlow have dependent components. In contrast,* $\mathcal{N}$ATTACK *perturbation components are independent.*

Table 1: Area under the receiver operating characteristic curve (AUROC) and accuracy of detecting adversarial examples generated by $\mathcal{N}$ATTACK [33] and AdvFlow (un. for un-trained and tr. for pre-trained NF) using LID [37], Mahalanobis [32], and Res-Flow [64] adversarial attack detectors. In each case, the classifier has a ResNet-34 [18] architecture.

| Data | Metric | AUROC(%) ↑ | | | Detection Acc.(%) ↑ | | |
|---|---|---|---|---|---|---|---|
| | Method | $\mathcal{N}$ATTACK | AdvFlow (un.) | AdvFlow (tr.) | $\mathcal{N}$ATTACK | AdvFlow (un.) | AdvFlow (tr.) |
| CIFAR-10 | LID [37] | 78.69 | 84.39 | **57.59** | 72.12 | 77.11 | **55.74** |
| | Mahalanobis [32] | 97.95 | 99.50 | **66.85** | 95.59 | 97.46 | **62.21** |
| | Res-Flow [64] | 97.90 | 99.40 | **67.03** | 94.55 | 97.21 | **62.60** |
| SVHN | LID [37] | **57.70** | 58.92 | 61.11 | **55.60** | 56.43 | 58.21 |
| | Mahalanobis [32] | 73.17 | 74.67 | **64.72** | 68.20 | 69.46 | **60.88** |
| | Res-Flow [64] | 69.70 | 74.86 | **64.68** | 64.53 | 68.41 | **61.13** |

## 4 Experimental Results

In this section, we present our experimental results. First, we see how the adversarial examples generated by the proposed model can successfully mislead adversarial example detectors. Then, we show the attack success rate and the number of queries required to attack both vanilla and defended models. Finally, we examine the transferability of the generated attacks between defended classifiers. To see the details of the experiments, please refer to Appendix B. Also, more simulation results can be found in Appendices C and D.

For each dataset, we pre-train a flow-based model and fix it across the experiments. To this end, we use a modified version of Real NVP [8] as introduced in [1], the details of which can be found in Appendix B.1. Once trained, we then try to attack target classifiers in a black-box setting using AdvFlow (Algorithm 1).

### 4.1 Detectability

One approach to defend pre-trained classifiers is to employ adversarial example detectors. This way, a detector is trained and put on top of the classifier. Before feeding inputs to the un-defended classifier, every input has to be checked whether it is adversarial or not. One common assumption among such detectors is that the adversaries come from a different distribution than the clean data [37, 32]. Thus, the performance of these detectors seems to be a suitable measure to quantify the success of our model in generating adversarial examples that have the same distribution as the original data. To this end, we choose LID [37], Mahalanobis [32], and Res-Flow [64] adversarial attack detectors to assess the performance of the proposed approach. We compare our results with $\mathcal{N}$ATTACK [33] *that also approaches the black-box adversarial attack from a distributional perspective* for a fair comparison. As an ablation study, we also consider the un-trained version of AdvFlows where the weights of the NF models are set randomly. This way, we can observe the effect of the clean data distribution in misleading adversarial example detectors more precisely. We first generate a set of adversarial examples alongside some noisy ones using the test set. Then, we use $10\%$ of the adversarial, noisy, and clean image data to train adversarial attack detectors. Details of our experiments in this section can be found in Appendix B.2.

We report the area under the receiver operating characteristic curve (AUROC) and the detection accuracy for each case in Table 1. As seen, in almost all the cases the selected adversarial detectors struggle to detect the attacks generated by AdvFlow in contrast to $\mathcal{N}$ATTACK. These results support our statement earlier about the distribution of the attacks being more similar to that of data, hence the failure of adversarial example detectors. Also, we see that pre-training the AdvFlow using clean data is crucial in fooling adversarial example detectors.

Finally, Figure 2 shows the relative change in the base distribution of the flow-based model for adversarial examples of Table 1. Interestingly, we see that AdvFlow adversaries are distinctively closer to the clean data compared to $\mathcal{N}$ATTACK [33]. These results highlight the need to reconsider the underlying assumption that adversaries come from a different distribution than the clean data. Also, it can motivate training classifiers that learn data distributions, as our results reveal this is not currently the case.

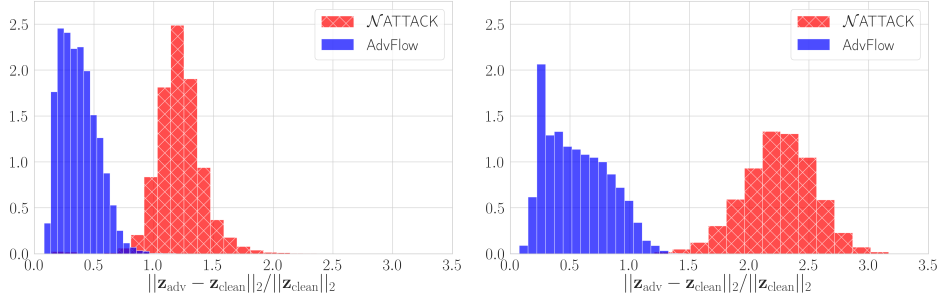

Figure 2: Relative change in the base distribution of the flow-based model for adversarial examples generated by AdvFlow and $\mathcal{N}$ATTACK for CIFAR-10 [30] (left) and SVHN [40] (right) classifiers of Table 1.

## 4.2  Success Rate and Number of Queries

Next, we investigate the performance of the proposed model in attacking vanilla and defended image classifiers. It was shown previously that $\mathcal{N}$ATTACK struggles to break into adversarially trained models more than any other defense [33]. Thus, we select some of the most recent defense techniques that are built upon adversarial training [38]. This selection also helps us in quantifying the attack transferability in our next experiment. Therefore, we select Free [48] and Fast [60] adversarial training, alongside adversarial training with auxiliary rotations [19] as the defense mechanisms that our classifiers employ. Note that these models are the most recent defenses built upon adversarial training. For a brief explanation of each one of these methods, please refer to Appendix B.3. We then train target classifiers on CIFAR-10 [30] and SVHN [40] datasets. The architecture that we use here is the well-known Wide-ResNet-32 [63] with width 10. We then try to attack these classifiers by generating adversarial examples on the test set. We compare our proposed model with bandits with time and data-dependent priors [23], $\mathcal{N}$ATTACK [33], and SimBA [16]. [4] To simulate a realistic environment, we set the maximum number of queries to $10,000$. Moreover, for $\mathcal{N}$ATTACK and AdvFlow we use a population size of 20. More details on the defense methods as well as attack hyperparameters can be found in Appendices B.3 and B.4.

Tables 2 and 3 show the success rate as well as the average and median number of queries required to successfully attack a vanilla/defended classifier. Also, Figure 3 in Appendix C shows the attack success rate for AdvFlow and $\mathcal{N}$ATTACK [33] versus the maximum number of queries for defended models. As can be seen, AdvFlows can improve upon the performance of $\mathcal{N}$ATTACK [33] in all of the defended models in terms of the number of queries and attack success rate. Also, it should be noted that although our performance on vanilla classifiers is worse than $\mathcal{N}$ATTACK [33], we are still generating adversaries that are not easily detectable by adversarial example detectors and come from a similar distribution to the clean data.

## 4.3  Transferability

Finally, we examine the transferability of the generated attacks for each of the classifiers in Table 2. In other words, we generate attacks using a substitute classifier, and then try to attack another target model. The results of this experiment are shown in Figure 4 of Appendix C. As seen, the generated attacks by AdvFlow transfer to other defended models easier than the vanilla one. This observation precisely matches our intuition about the mechanics of AdvFlow. More specifically, we know that in AdvFlow the model is learning a distribution that is more expressive than the one used by $\mathcal{N}$ATTACK. Also, we have seen in Section 3.4.2 that the perturbations generated by AdvFlow have dependent elements in contrast to $\mathcal{N}$ATTACK. As a result, AdvFlow learns to attack classifiers using higher-level features (Figure 1). Thus, since vanilla classifiers use different features for classification than the defended ones, AdvFlows are less transferable from defended models to vanilla ones. In contrast,

Table 2: Attack success rate of black-box adversarial attacks on CIFAR-10 [30] and SVHN [40] Wide-ResNet-32 [63] classifiers. All attacks are with respect to $\ell_\infty$ norm with $\epsilon_{\max} = 8/255$.

| Data | Defense | Acc.(%) | Success Rate(%) ↑ | | | |
|---|---|---|---|---|---|---|
| | | | Bandits[23] | $\mathcal{N}$ATTACK [33] | SimBA [16] | AdvFlow |
| CIFAR-10 | Vanilla [63] | 91.77 | 98.81 | **100** | 99.99 | 99.42 |
| | FreeAdv [48] | 81.29 | 37.12 | 38.97 | 35.52 | **41.21** |
| | FastAdv [60] | 86.33 | 36.60 | 36.90 | 35.07 | **40.22** |
| | RotNetAdv [19] | 86.58 | 37.73 | 38.04 | 35.63 | **40.67** |
| SVHN | Vanilla [63] | 96.45 | 87.84 | **98.76** | 97.26 | 90.31 |
| | FreeAdv [48] | 86.47 | 49.64 | 50.28 | 46.28 | **50.76** |
| | FastAdv [60] | 93.90 | 40.43 | 35.42 | 36.19 | **41.49** |
| | RotNetAdv [19] | 90.33 | 43.47 | 41.49 | 39.01 | **44.22** |

Table 3: Average (median) of the number of queries needed to generate an adversarial example for CIFAR-10 [30] and SVHN [40] Wide-ResNet-32 [63] classifiers of Table 2. For a fair comparison, we first find the samples where all the attack methods are successful, and then compute the average (median) of queries for these samples. Note that for $\mathcal{N}$ATTACK and AdvFlow we check whether we arrived at an adversarial point every 200 queries, and hence, the medians are a multiples of 200.

| Data | Defense | Query Average (Median) on Mutually Successful Attacks ↓ | | | |
|---|---|---|---|---|---|
| | | Bandits[23] | $\mathcal{N}$ATTACK [33] | SimBA [16] | AdvFlow |
| CIFAR-10 | Vanilla [63] | 552.69 (182) | **237.58** (200) | 237.70 (**126**) | 949.31 (400) |
| | FreeAdv [48] | 1062.7 (354) | 874.91 (400) | 463.09 (244) | **421.63** (**200**) |
| | FastAdv [60] | 1065.92 (358) | 973.05 (400) | **428.81** (234) | 436.8 (**200**) |
| | RotNetAdv [19] | 1085.43 (408) | 941.67 (400) | 471.99 (259) | **424.95** (**200**) |
| SVHN | Vanilla [63] | 1750.65 (1128) | 408.75 (200) | **202.07** (**107**) | 1572.24 (600) |
| | FreeAdv [48] | 819.98 (250) | 903.12 (400) | **365.42** (216) | 692.73 (**200**) |
| | FastAdv [60] | 755.23 (284) | 1243.38 (600) | **307.73** (216) | 526.37 (**200**) |
| | RotNetAdv [19] | 663.07 (202) | 756.48 (400) | **319.93** (**186**) | 480.02 (200) |

the expressiveness of AdvFlows enables the attacks to be transferred more successfully between adversarially trained classifiers, and from vanilla to defended ones.

## 5   Conclusion and Future Directions

In this paper, we introduced AdvFlow: a novel adversarial attack model that utilizes the capacity of normalizing flows in representing data distributions. We saw that the adversarial perturbations generated by the proposed approach can be approximated using normal distributions with dependent components. In this sense, $\mathcal{N}$ATTACK [33] cannot generate such adversaries. As a result, AdvFlows are less conspicuous to adversarial example detectors in contrast to their $\mathcal{N}$ATTACK [33] counterpart. This success is due to AdvFlow being pre-trained on the data distribution, resulting in adversaries that look like the clean data. We also saw the capability of the proposed method in improving the performance of bandits [23], $\mathcal{N}$ATTACK [33], and SimBA [16] on adversarially trained classifiers. This improvement is in terms of both attack success rate and the number of queries.

Flow-based modeling is an active area of research. There are numerous extensions to the current work that can be investigated upon successful expansion of normalizing flow models in their range and power. For example, while $\mathcal{N}$ATTACK [33] and other similar approaches [22, 23] are specifically designed for use on image data, the current work can potentially be expanded to entail other forms of data such as graphs [35, 49]. Also, since normalizing flows can effectively model probability distributions, finding the distribution of well-known perturbations may lead to increasing classifier robustness against adversarial examples. We hope that this work can provide a stepping stone to exploiting such powerful models for adversarial machine learning.

## Broader Impact

In this paper, we introduce a novel adversarial attack algorithm called AdvFlow. It uses pre-trained normalizing flows to generate adversarial examples. This study is crucial as it indicates the vulnerability of deep neural network (DNN) classifiers to adversarial attacks.

More precisely, our study reveals that the common assumption made by adversarial example detectors (such as the Mahalanobis detector [32]) that the adversaries come from a different distribution than the data may not be an accurate one. In particular, we show that we can generate adversaries that come from a close distribution to the data, yet they intend to mislead the classifier decision. Thus, we emphasize that adversarial example detectors need to adjust their assumption about the distribution of adversaries before being deployed in real-world situations.

Furthermore, since our adversarial examples are closely related to the data distribution, our method shows that DNN classifiers are not learning to classify the data based on their underlying distribution. Otherwise, they would have resisted the attacks generated by AdvFlow. Thus, it can bring the attention of the machine learning community to training their DNN classifiers in a distributional sense.

All in all, we pinpoint a failure of DNN classifiers to the rest of the community so that they can become familiar with the limitations of the status-quo. This study, and similar ones, could raise awareness among researchers about the real-world pitfalls of DNN classifiers, with the aim of consolidating them against such threats in the future.

## Acknowledgments and Disclosure of Funding

We would like to thank the reviewers for their valuable feedback on our work, helping us to improve the final manuscript. We also would like to thank the authors and maintainers of PyTorch [43], NumPy [17], and Matplotlib [21].

This research was undertaken using the LIEF HPC-GPGPU Facility hosted at the University of Melbourne. This Facility was established with the assistance of LIEF Grant LE170100200.

## Footnotes

[1]Note that although we are defining our objective function $\mathcal{L}(\mathbf{x}')$ for un-targeted adversarial attacks, it can be easily modified to targeted attacks. To this end, it suffices to replace $\max_{c \neq y} \log \mathcal{C}(\mathbf{x}')_c$ in Eq. (1) with $\log \mathcal{C}(\mathbf{x}')_t$, where $t$ shows the target class output. In this paper, we only consider un-targeted attacks.

[2]Note that this update procedure is not what NES precisely stands for. It is rather a *canonical gradient search algorithm* as is called by Wierstra et al. [58], which only makes use of a *vanilla* gradient [59] for evolution strategies. In fact, the *natural* term in *natural evolution strategies* represents an update of the form $\boldsymbol{\psi} \leftarrow \boldsymbol{\psi} - \alpha\tilde{\nabla}_{\boldsymbol{\psi}} J(\boldsymbol{\psi})$, where $\tilde{\nabla}_{\boldsymbol{\psi}} J(\boldsymbol{\psi}) = \mathbf{F}^{-1}\nabla_{\boldsymbol{\psi}} J(\boldsymbol{\psi})$ is called the *natural gradient*. Here, the matrix $\mathbf{F}$ is the *Fischer information matrix* of the search distribution $p(\mathbf{x}'|\boldsymbol{\psi})$. However, since NES in the adversarial learning literature [22, 23, 33] points to Eq. (8), we use the same convention here.

[3]Indeed, we can also optimize $\sigma$ alongside $\boldsymbol{\mu}$ to enhance the attack strength. However, since $\mathcal{N}$ATTACK [33] only optimizes $\boldsymbol{\mu}$, we also stick with the same setting.

[4]Note that SimBA [16] is originally designed for efficient $\ell_2$ attacks, and it may not use the entire $10,000$ query quota for small images. Anyways, we included SimBA [16] in the paper as per one of the reviewers' suggestions.

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
