[Supplementary Material]

# Supplementary Materials

This supplementary document includes the following content to support the material presented in the paper:

- In Section A, we present the "log-likelihood trick" and the proof to our lemma and corollary.
- In Section B, we give the implementation details of our algorithm and experiments. Besides introducing the flow-based model architecture, we present a detailed explanation of classifier architectures and defense mechanisms used to evaluate our method. The set of hyperparameters used in each defense and attack model is also given.
- In Section C, we present an extended version of our simulation results. Moreover, we investigate the effect of the training data on our algorithm's performance.
- In Section D, we see important extensions to the current work. These extensions include our solution to high-resolution images and un-trained AdvFlow.

## A  Mathematical Details

### A.1  The Log-likelihood Trick

Here we provide the complete proof of the "log-likelihood trick" as presented in [58]:

$$
\begin{aligned}
\nabla_{\boldsymbol{\psi}} J(\boldsymbol{\psi}) &= \nabla_{\boldsymbol{\psi}} \mathbb{E}_{p(\mathbf{x}'|\boldsymbol{\psi})} \left[ \mathcal{L}(\mathbf{x}') \right] \\
&= \nabla_{\boldsymbol{\psi}} \int \mathcal{L}(\mathbf{x}') p(\mathbf{x}'|\boldsymbol{\psi}) \mathrm{d}\mathbf{x}' \\
&= \int \mathcal{L}(\mathbf{x}') \nabla_{\boldsymbol{\psi}} p(\mathbf{x}'|\boldsymbol{\psi}) \mathrm{d}\mathbf{x}' \\
&= \int \mathcal{L}(\mathbf{x}') \frac{\nabla_{\boldsymbol{\psi}} p(\mathbf{x}'|\boldsymbol{\psi})}{p(\mathbf{x}'|\boldsymbol{\psi})} p(\mathbf{x}'|\boldsymbol{\psi}) \mathrm{d}\mathbf{x}' \\
&= \int \mathcal{L}(\mathbf{x}') \nabla_{\boldsymbol{\psi}} \log \left( p(\mathbf{x}'|\boldsymbol{\psi}) \right) p(\mathbf{x}'|\boldsymbol{\psi}) \mathrm{d}\mathbf{x}' \\
&= \mathbb{E}_{p(\mathbf{x}'|\boldsymbol{\psi})} \left[ \mathcal{L}(\mathbf{x}') \nabla_{\boldsymbol{\psi}} \log \left( p(\mathbf{x}'|\boldsymbol{\psi}) \right) \right].
\end{aligned}
$$

### A.2  Proofs

**Lemma A.1.** *Let* $\mathbf{f}(\mathbf{x})$ *be an invertible, differentiable function. For a small perturbation* $\boldsymbol{\delta}_z$ *we have*

$$
\boldsymbol{\delta} = \mathbf{f}\left(\mathbf{f}^{-1}(\mathbf{x}) + \boldsymbol{\delta}_z\right) - \mathbf{x} \approx \left(\nabla \mathbf{f}^{-1}(\mathbf{x})\right)^{-1} \boldsymbol{\delta}_z.
$$

*Proof.* By the first-order Taylor series for $\mathbf{f}\left(\cdot\right) : \mathbb{R}^d \to \mathbb{R}^d$ we have

$$
\mathbf{f}(\mathbf{z} + \boldsymbol{\delta}_z) \approx \mathbf{f}(\mathbf{z}) + \nabla \mathbf{f}(\mathbf{z}) \boldsymbol{\delta}_z,
$$

where $\nabla \mathbf{f}(\mathbf{z})$ is the $d \times d$ Jacobian matrix of the function $\mathbf{f}(\cdot)$. Now, by substituting $\mathbf{z} = \mathbf{f}^{-1}(\mathbf{x})$, and using the *inverse function theorem* [45], we can write

$$
\mathbf{f}(\mathbf{f}^{-1}(\mathbf{x}) + \boldsymbol{\delta}_z) \approx \mathbf{f}(\mathbf{f}^{-1}(\mathbf{x})) + \left(\nabla \mathbf{f}^{-1}(\mathbf{x})\right)^{-1} \boldsymbol{\delta}_z
$$

which gives us the result immediately. $\square$

**Corollary A.1.1.** *The adversarial perturbations generated by AdvFlow have dependent components. In contrast,* $\mathcal{N}$ ATTACK *perturbation components are independent.*

*Proof.* By Lemma 3.1 we know that

$$
\boldsymbol{\delta} \approx \left(\nabla \mathbf{f}^{-1}(\mathbf{x})\right)^{-1} \boldsymbol{\delta}_z,
$$

where $\boldsymbol{\delta}_z$ is distributed according to an isometric normal distribution. For $\mathcal{N}$ ATTACK, we have $\mathbf{f}(\mathbf{z}) = \frac{1}{2}\left(\tanh(\mathbf{z}) + 1\right)$. Thus, it can be shown that $\nabla \mathbf{f}^{-1}(\mathbf{x}) = \mathrm{diag}\left(2\mathbf{x} \odot (1 - \mathbf{x})\right)$. As a result,

the random vector $\boldsymbol{\delta}$ will be still normally distributed with a diagonal covariance matrix, and hence, have independent components. In contrast, we know that for an effective flow-based model $\nabla \mathbf{f}^{-1}(\mathbf{x})$ is not always diagonal. Otherwise, this means that our NF is simply a data-independent affine transformation. For example, in Real NVP [8] which we use, this matrix is a product of lower and upper triangular matrices. Hence, for a normalizing flow model $\mathbf{f}(\cdot)$ we have non-diagonal $\nabla \mathbf{f}^{-1}(\mathbf{x})$. Thus, it will make the random variable $\boldsymbol{\delta}$ normal with correlated (dependent) components. $\quad\square$

# B   Implementation Details

In this section, we present the implementation details of our algorithm and experiments. Note that all of the experiments were run using a single NVIDIA Tesla V100-SXM2-16GB GPU.

## B.1   Normalizing Flows

For the flow-based models used in AdvFlow, we implement Real NVP [8] by using the framework of Ardizzone et al. [1].[5] In particular, let $\mathbf{z} = [\mathbf{z}_1, \mathbf{z}_2]$ denote the input to one layer of a normalizing flow transformation. If we denote the output of this layer by $\mathbf{x} = [\mathbf{x}_1, \mathbf{x}_2]$, then the Real NVP transformation between the input and output of this particular layer can be written as:

$$\mathbf{x}_1 = \mathbf{z}_1 \odot \exp\big(\mathbf{s}_1\left(\mathbf{z}_2\right)\big) + \mathbf{t}_1\left(\mathbf{z}_2\right)$$
$$\mathbf{x}_2 = \mathbf{z}_2 \odot \exp\big(\mathbf{s}_2\left(\mathbf{x}_1\right)\big) + \mathbf{t}_2\left(\mathbf{x}_1\right),$$

where $\odot$ is an element-wise multiplication. The functions $\mathbf{s}_{1,2}(\cdot)$ and $\mathbf{t}_{1,2}(\cdot)$ are called the scaling and translation functions. Since the invertibility of the transformation does not depend on these functions, they are implemented using ordinary neural networks. To help with the stability of the transformation, Ardizzone et al. [1] suggest using a soft-clamp before passing the output of scaling networks $\mathbf{s}_{1,2}(\cdot)$ to exponential function. This soft-clamp function is implemented by

$$s_{\text{clamp}} = \frac{2\alpha}{\pi} \arctan\left(\frac{s}{\alpha}\right),$$

where $\alpha$ is a hyperparameter that controls the amount of softening. In our experiments, we set $\alpha = 1.5$. Moreover, at the end of each layer of transformation, we permute the output so that we end up getting different partitions of the data as $\mathbf{z}_1$ and $\mathbf{z}_2$. The pattern by which the data is permuted is set at random at the beginning of the training process and kept fixed onwards.

After passing the data through some high-resolution transformations, we downsample it using i-RevNet downsamplers [24]. Specifically, the high-resolution input is downsampled so that each one of them constitutes a different channel of the low-resolution data.

To help the normalizing flow model learn useful features, we use a fixed $1 \times 1$ convolution at the beginning of each low-resolution layer. This adjustment is done with the same spirit as in Glow [27]. However, instead of having a trainable $1 \times 1$ convolution, here we initialize them at the beginning of the training and keep them fixed afterward.

Finally, we used a multi-scale structure [8] to reduce the computational complexity of the flow-based model. Specifically, we pass the input through several layers of invertible transformations constructed using convolutional neural networks as $\mathbf{s}_{1,2}(\cdot)$ and $\mathbf{t}_{1,2}(\cdot)$. Then, we send three-quarters of the data directly to the ultimate output. The rest goes through other rounds of mappings, which use fully-connected networks. This way, one can reduce the computational burden of flow-based models as they keep the data dimension fixed.

For training, we use an Adam [26] optimizer with weight decay $10^{-5}$. Besides, we set the learning rate according to an exponential scheduler starting from $10^{-4}$ and ending to $10^{-6}$. Also, to dequantize the image pixels, we add a small Gaussian noise with $\sigma = 0.02$ to the pictures. Table 4 summarizes the hyperparameters and architecture of the flow-based model used in AdvFlow.

## B.2   Adversarial Example Detectors

In this section, we provide the details of the LID [37], Mahalanobis [32], and ResFlow [64] adversarial example detectors. All of the methods use logistic regression as their classifier, and the way that they

Table 4: Hyperparameter and architecture details for normalizing flow part of AdvFlow.

| | |
|---|---|
| Optimizer | Adam |
| Scheduler | Exponential |
| Initial lr | $10^{-4}$ |
| Final lr | $10^{-6}$ |
| Batch Size | 64 |
| Epochs | 350 |
| Added Noise Std. | 0.02 |
| Multi-scale Levels | 2 |
| Each Level Network Type | CNN-FC |
| High-res Transformation Blocks | 4 |
| Low-res Transformation Blocks | 6 |
| FC Transformation Blocks | 6 |
| $\alpha$ (clamping hyperparameter) | 1.5 |
| CNN Layers Hidden Channels | 128 |
| FC Layers Internal Width | 128 |
| Activation Function | Leaky ReLU |
| Leaky Slope | 0.1 |

construct their training and evaluation sets is the same. The only difference among these methods is the way each one extracts their features, which we review below. The training set used to train the logistic regression classifier consists of three types of data: clean, noisy, and adversarial. We take the test portion of each target dataset and add a slight noise to them to make the noisy data. The clean and noisy data are going to be used as the positive samples of the logistic regression. For the adversarial part of the data, we then use a nominated adversarial attack method and generate adversarial examples that are later used as negative samples of the logistic regression. After constructing the entire dataset, 10% of it is used as the logistic regression training set, and the rest for evaluation. Also, the hyperparameters of the detectors are set using nested cross-validation.

**LID Detectors.** Ma et al. [37] use the concept of *Local Intrinsic Dimensionality* (LID) to charac-terize adversarial subspaces. It is argued that for a data point that resides on some high-dimensional submanifold, its adversarially generated sample is likely to lie outside this submanifold. As such, Ma et al. [37] argue that the intrinsic dimensionality of the adversarial examples in a local neighborhood is going to be higher than the clean or noisy data (see Figure 1 of [37]). Thus, LID can be a good measure for differentiating adversarial examples from clean data. Ma et al. [37] then estimate the LID measure for mini-batches of data using the extreme value theory. To this end, they extract features of the input images using a DNN classifier. They then compute the LID score for all these features across the training and evaluation sets. After extracting these scores for all the data, they train and evaluate the logistic regression classifier as described above. Here, we use the PyTorch implementation of LID detectors given by Lee et al. [32].

**Mahalanobis Detectors.** Lee et al. [32] propose an adversarial example detector based on a Mahalanobis distance-based confidence score. To this end, the authors extract features from different hidden layers of a nominated DNN classifier. Assuming that these features are distributed according to class-conditional Gaussian densities, the detector aims at estimating the mean and covariance matrix associated with each one of the features across the training set. These densities are then used to train the logistic regression classifier based on the Mahalanobis distance confidence score between a given image feature and its closest distribution. In this paper, we use the official implementation of the Mahalanobis adversarial example detector available online.[6] For more information about these detectors, see [32].

**ResFlow Detectors.** Zisselman and Tamar [64] generalize the Mahalanobis detectors [32] using normalizing flows. It is first argued that modeling the activation distributions as Gaussian densities may not be accurate. To find a better non-Gaussian distribution, Zisselman and Tamar [64] exploit flow-based models to construct an architecture they call Residual Flow (ResFlow). The same

procedure as in the Mahalanobis detectors is then utilized to extract features that are later used to train the logistic regression detectors. We use the official PyTorch implementation of ResFlow available online.[7] Note that in the original paper, ResFlows are only used for out-of-distribution detection. For our purposes, we generalized their implementation to adversarial example detection using the Mahalanobis detector implementation.

## B.3 Defense Methods

In this section, we briefly review the defense techniques used in our experiments. We will then present the set of parameters used in the training of each one of the classifiers. Note that we only utilized these methods for the sake of evaluating our attack models, and our results cannot be regarded as a close case-study, or comparison, of the nominated defense methods.

### B.3.1 Review of Defense Methods

**Adversarial Training.** Adversarial training [38] is a method to train robust classifiers. To achieve robustness, this method tries to incorporate adversarial examples into the training process. In particular, adversarial training aims at minimizing the following objective function for classifier $\mathcal{C}(\cdot)$ with parameters $\boldsymbol{\theta}$:

$$\min_{\boldsymbol{\theta}} \sum_i \max_{\|\boldsymbol{\delta}\| \leq \epsilon} \ell\big(\mathcal{C}(\mathbf{x}_i + \boldsymbol{\delta}), y_i\big). \tag{12}$$

Here, $\mathbf{x}_i$ and $y_i$ are the training examples and their associated correct labels. Also, $\ell(\cdot)$ is an appropriate cost function for classifiers, such as the standard cross-entropy loss. The inner maximization objective in Eq. (12) is the cost function used to generate adversarial examples. Thus, we can interpret Eq. (12) as training a model that can predict the labels correctly, even in the presence of additive perturbations. However, finding the exact solution to the inner optimization problem is not straightforward, and in most real-world cases cannot be done efficiently [29]. To circumvent this problem, Madry et al. [38] proposed approximately solving it by using a Projected Gradient Descent algorithm. This method is widely known as adversarial training.

**Adversarial Training for Free.** The main disadvantage of adversarial training, as proposed in [38], is that solving the inner optimization problem makes the algorithm much slower than standard classifier training. This problem arises because solving the inner maximization objective requires back-propagating through the DNN. To address this issue, Shafahi et al. [48] exploits a Fast Gradient Sign Method (FGSM) [14] with step-size $\epsilon$ to compute an approximate solution to the inner maximization objective and then update the DNN parameters. This procedure is repeated $m$ times on the same minibatch. Finally, the total number of epochs is divided by a factor of $m$ to account for repeated minibatch training. We use the PyTorch code available on the official repository of the *free adversarial training* to train our classifiers.[8]

**Fast Adversarial Training.** To make adversarial training even faster, Wong et al. [60] came up with a method called "fast" adversarial training. In this approach, they combine FGSM adversarial training with the idea of random initialization to train robust DNN classifiers. To make the proposed algorithm even faster, Wong et al. [60] also utilize several fast training techniques (such as cyclic learning rate and mixed-precision arithmetic) from DAWNBench competition [6]. In this paper, we replace the FGSM adversarial training with PGD. However, we still use the cyclic learning rate and mixed-precision arithmetic. For this method, we used the official PyTorch code available online.[9]

**Adversarial Training with Auxiliary Rotations.** Gidaris et al. [13] showed that Convolutional Neural Networks can learn useful image features in an unsupervised fashion by predicting the amount of rotation applied to a given image. Throughout their experiments, they observed that these features can improve classification performance. Motivated by these observations, Hendrycks et al. [19] suggest exploiting the idea of self-supervised feature learning to improve the robustness of classifiers against adversaries. Specifically, it is proposed to train a so-called "head" alongside the original classifier. This auxiliary head takes the penultimate features of the classifier and aims at predicting

the amount of rotation applied to an image from four possible angles ($0°$, $90°$, $180°$, or $270°$). It was shown that this simple addition can improve the performance of adversarially trained classifiers. To train our models, we make use of the PyTorch code for *adversarial learning with auxiliary rotations* available online.[10]

### B.3.2   Hyperparameters of Defense Methods

Tables 5 and 6 summarize the hyperparameters used for training our defended classifiers.

Table 5: Hyperparameters of defense methods for training CIFAR-10 [30] classifiers. Numbers 1 and 2 correspond to Wide-ResNet-32 [63] and ResNet-50 [18] architectures, respectively.

| Classifier | Free-1 | Free-2 | Fast-1 | Fast-2 | RotNet-1 | RotNet-2 |
|---|---|---|---|---|---|---|
| Optimizer | SGD | SGD | SGD | SGD | SGD | SGD |
| lr | 0.1 | 0.005 | 0.21 | 0.21 | 0.1 | 0.1 |
| Momentum | 0.9 | 0.9 | 0.9 | 0.9 | 0.9 | 0.9 |
| Weight Decay | 0.0002 | 0.0002 | 0.0005 | 0.0005 | 0.0005 | 0.0005 |
| Nesterov | N | N | N | N | Y | Y |
| Batch Size | 128 | 64 | 64 | 128 | 128 | 128 |
| Epochs | 125 | 100 | 100 | 100 | 100 | 100 |
| Inner Optimization | FGSM | FGSM | PGD | PGD | PGD | PGD |
| $\epsilon$ | 8/255 | 8/255 | 8/255 | 8/255 | 8/255 | 8/255 |
| Step Size | 8/255 | 8/255 | 2/255 | 2/255 | 2/255 | 2/255 |
| Number of Steps (Repeats) | 8 | 8 | 5 | 5 | 10 | 10 |

Table 6: Hyperparameters of defense methods for training SVHN [40] classifiers. Numbers 1 and 2 correspond to Wide-ResNet-32 [63] and ResNet-50 [18] architectures, respectively.

| Classifier | Free-1 | Free-2 | Fast-1 | Fast-2 | RotNet-1 | RotNet-2 |
|---|---|---|---|---|---|---|
| Optimizer | SGD | SGD | SGD | SGD | SGD | SGD |
| lr | 0.0001 | 0.01 | 0.21 | 0.21 | 0.1 | 0.1 |
| Momentum | 0.9 | 0.9 | 0.9 | 0.9 | 0.9 | 0.9 |
| Weight Decay | 0.0002 | 0.0002 | 0.0005 | 0.0005 | 0.0005 | 0.0005 |
| Nesterov | N | N | N | N | Y | Y |
| Batch Size | 128 | 128 | 64 | 128 | 128 | 128 |
| Epochs | 100 | 100 | 100 | 100 | 100 | 100 |
| Inner Optimization | FGSM | FGSM | PGD | PGD | PGD | PGD |
| $\epsilon$ | 8/255 | 8/255 | 8/255 | 8/255 | 8/255 | 8/255 |
| Step Size | 8/255 | 8/255 | 2/255 | 2/255 | 2/255 | 2/255 |
| Number of Steps (Repeats) | 8 | 8 | 5 | 5 | 10 | 10 |

### B.4   Hyperparameters of Attack Methods

In this part, we present the set of hyperparameters used for each attack method. For $\mathcal{N}$ATTACK [33] and AdvFlow, we tune the hyperparameters on a development set so that they result in the best performance for an un-defended CIFAR-10 classifier. In the case of bandits with time and data-dependent priors [23], we use two sets of hyperparameters tuned for these methods. For the vanilla classifiers we use the hyperparameters set in [23], while for defended classifiers we use those set in [39]. For SimBA [16], we used the hyperparameters set in the official repository. We only changed the stride from 7 to 6 to allow for the correct computation of block reordering. Once set, we keep the hyperparameters fixed throughout the rest of experiments. Tables 7-10 summarize the hyperparameters used for each attack method in our experiments.

Table 7: Hyperparameters of bandits with time and data-dependent priors [23].

| Hyperparameter | Vanilla | Defended |
|---|---|---|
| OCO learning rate | 100 | 0.1 |
| Image learning rate | 0.01 | 0.01 |
| Bandit exploration | 0.1 | 0.1 |
| Finite difference probe | 0.1 | 0.1 |
| Tile size | $(6\text{px})^2$ | $(4\text{px})^2$ |

Table 8: Hyperparameters of $\mathcal{N}$ATTACK [33].

| Hyperparameter | Value |
|---|---|
| $\sigma$ (noise std.) | 0.1 |
| Sample size | 20 |
| Learning rate | 0.02 |
| Maximum iteration | 500 |

Table 9: Hyperparameters of SimBA [16].

| Hyperparameter | Value |
|---|---|
| $\epsilon$ | 0.2 |
| Freq. Dimensionality | 14 |
| Order | Strided |
| Stride | 6 |

Table 10: Hyperparameters of AdvFlow (ours).

| Hyperparameter | Value |
|---|---|
| $\sigma$ (noise std.) | 0.1 |
| Sample size | 20 |
| Learning rate | 0.02 |
| Maximum iteration | 500 |

# C Extended Experimental Results

In this section, we present an extended version of our experimental results.

## C.1 Table of Attack Success Rate and Number of Queries

Table 11 presents attack success rate, as well as average and median of the number of queries for AdvFlow alongside bandits [23] and $\mathcal{N}$ ATTACK [33]. In each case, we have also shown the clean data accuracy and success rate of the white-box PGD-100 attack for reference. Details of classifier training and defense mechanism can be found in Appendix B.3. As can be seen, when it comes to attacking defended models, AdvFlow can outperform the baselines in both the number of queries and attack success rate.

## C.2 Success Rate vs. Number of Queries

Figure 3 shows the success rate of AdvFlow and $\mathcal{N}$ ATTACK [33] as a function of the maximum number of queries for defended models. As can be seen, given a fixed number of queries, AdvFlow can generate more successful attacks.

## C.3 Confusion Matrices of Transferability

Figure 4 shows the transferability rate of generated attacks to various classifiers. Each entry shows the success rate of adversarial examples intended to attack the row-wise classifier in attacking the column-wise classifier. There are a few points worth mentioning regarding these results:

- AdvFlow attacks are more transferable between defended models than vanilla to defended models. We argue that the underlying reason is the fact that AdvFlow learns a higher-level perturbation to attack DNNs. As a result, since vanilla classifiers use different features than the defended ones, they are less adaptable to attack defended classifiers. In contrast, since $\mathcal{N}$ ATTACK acts on a pixel level, they are less susceptible to this issue.

- Generally, adversarial examples generated by AdvFlow are more transferable between different architectures than $\mathcal{N}$ ATTACK. The same argument as in our previous point applies here.

- Transferability of black-box attacks is not as important as in the white-box setting. The reason is that in the case of black-box attacks, since no assumption is made about the model architecture, we can try to generate new adversarial examples to attack a new target classifier. However, for white-box attacks, transferability is somehow related to their success in attacking previously unseen target networks. Thus, it is essential to have a high rate of transferability if a white-box attack is meant to be deployed in real-world situations where often, we do not have any access to internal nodes of a classifier.

## C.4 Samples of Adversarial Examples

Figure 5 shows samples of adversarial examples generated by AdvFlow and $\mathcal{N}$ ATTACK [33], intended to attack a vanilla Wide-ResNet-32 [63]. As the images show, AdvFlow can generate adversarial perturbations that often take the shape of the original data. This property makes AdvFlows less detectable to adversarial example detectors. In contrast, it is clear that the perturbations generated by $\mathcal{N}$ ATTACK come from a different distribution than the data itself. As a result, they can be detected easily by adversarial example detectors.

Table 11: Attack success rate, average and median of the number of queries to generate an adversarial example for CIFAR-10 [30] and SVHN [40]. For a fair comparison, we first find the samples where all the attack methods are successful, and then compute the average (median) of queries for these samples. Note that for $\mathcal{N}$ATTACK and AdvFlow we check whether we arrived at an adversarial point every 200 queries, and hence, the medians are multiples of 200. Clean data accuracy and PGD-100 attack success rate are also shown for reference. All attacks are with respect to $\ell_\infty$ norm with $\epsilon_{max} = 8/255$.

| Arch | Data | Defense | Attack Clean Acc.(%) | PGD-100 Success Rate(%)↑ | Bandits [23] / $\mathcal{N}$ATTACK [33] / SimBA [16] / AdvFlow (ours) | | |
|---|---|---|---|---|---|---|---|
| | | | | | Success Rate(%)↑ | Avg. of Queries↓ | Med. of Queries↓ |
| WideResNet32 [63] | CIFAR-10 | Vanilla | 91.77 | 100 | 98.81 / **100** / 99.99 / 99.42 | 552.69 / **237.58** / 237.70 / 949.31 | 182 / 200 / **126** / 400 |
| | | FreeAdv [48] | 81.29 | 47.52 | 37.12 / 38.97 / 35.52 / **41.21** | 1062.70 / 874.91 / 463.09 / **421.63** | 354 / 400 / 244 / **200** |
| | | FastAdv [60] | 86.33 | 46.37 | 36.60 / 36.90 / 35.07 / **40.22** | 1065.92 / 973.05 / **428.81** / 436.80 | 358 / 400 / 234 / **200** |
| | | RotNetAdv [19] | 86.58 | 46.59 | 37.73 / 38.04 / 35.63 / **40.67** | 1085.43 / 941.67 / 471.99 / **424.95** | 408 / 400 / 259 / **200** |
| | SVHN | Vanilla | 96.45 | 99.81 | 87.84 / **98.76** / 97.26 / 90.31 | 1750.65 / 408.75 / **202.07** / 1572.24 | 1128 / 200 / **107** / 600 |
| | | FreeAdv [48] | 86.47 | 57.22 | 49.64 / 50.28 / 46.28 / **50.76** | 819.98 / 903.12 / **365.42** / 692.73 | 250 / 400 / 216 / **200** |
| | | FastAdv [60] | 93.90 | 46.76 | 40.43 / 35.42 / 36.19 / **41.49** | 755.23 / 1243.38 / **307.73** / 526.37 | 284 / 600 / 216 / **200** |
| | | RotNetAdv [19] | 90.33 | 48.67 | 43.47 / 41.49 / 39.01 / **44.22** | 663.07 / 756.48 / 319.93 / 480.02 | 202 / 400 / **186** / 200 |
| ResNet50 [18] | CIFAR-10 | Vanilla | 91.75 | 100 | 96.75 / 99.85 / **99.96** / 99.37 | 795.28 / **252.13** / 286.05 / 1051.18 | 280 / 200 / **163** / 600 |
| | | FreeAdv [48] | 75.17 | 54.54 | 45.64 / 46.49 / 43.14 / **49.46** | 842.56 / 836.81 / 383.56 / **371.81** | 248 / 400 / 206 / **200** |
| | | FastAdv [60] | 79.09 | 53.45 | 45.20 / 45.19 / 43.57 / **49.08** | 891.54 / 901.44 / 374.58 / **359.21** | 248 / 400 / 184 / **200** |
| | | RotNetAdv [19] | 76.39 | 52.04 | 45.80 / 46.41 / 42.65 / **50.10** | 826.60 / 774.24 / 376.30 / **292.74** | 232 / 400 / 184 / **200** |
| | SVHN | Vanilla | 96.23 | 99.38 | 92.63 / **96.73** / 93.14 / 83.67 | 1338.30 / 487.32 / **250.02** / 1749.48 | 852 / 200 / **126** / 800 |
| | | FreeAdv [48] | 87.67 | 46.50 | 42.27 / 43.99 / 39.83 / **44.66** | 793.30 / 703.76 / **327.30** / 565.2 | **198** / 400 / 207 / 200 |
| | | FastAdv [60] | 92.67 | 50.25 | 43.26 / 36.99 / 38.98 / **45.11** | 739.40 / 1255.24 / **286.71** / 436.83 | 294 / 600 / 202 / **200** |
| | | RotNetAdv [19] | 90.15 | 48.30 | 43.17 / 40.37 / 39.00 / **43.96** | 660.81 / 891.44 / 312.47 / 497.74 | **190** / 400 / 195 / 200 |

Figure 3: Success rate *vs.* maximum number of queries to attack CIFAR-10 [30] and SVHN [40] classifiers with Wide-ResNet-32 [63] and ResNet-50 [18] architectures.

**CIFAR-10 ($\mathcal{N}$Attack)**

| | Vanilla-1 | Free-1 | Fast-1 | RotNet-1 | Vanilla-2 | Free-2 | Fast-2 | RotNet-2 |
|---|---|---|---|---|---|---|---|---|
| Vanilla-1 | 100 | 1.03 | 0.62 | 0.76 | 13.5 | 1.07 | 1.21 | 1.16 |
| Free-1 | 22.32 | 100 | 13.04 | 10.44 | 22.67 | 25.81 | 20.78 | 19.23 |
| Fast-1 | 23.23 | 19.24 | 100 | 15.87 | 22.14 | 18.72 | 22.31 | 20.89 |
| RotNet-1 | 19.24 | 14.13 | 13.47 | 100 | 16.77 | 13.06 | 17.03 | 17.08 |
| Vanilla-2 | 19.15 | 1.18 | 0.6 | 0.71 | 100 | 1.2 | 1.2 | 1.33 |
| Free-2 | 12.87 | 10.15 | 5.2 | 4.26 | 13.3 | 100 | 11.55 | 13.08 |
| Fast-2 | 15 | 10.33 | 6.05 | 5.54 | 13.33 | 14.05 | 100 | 16.98 |
| RotNet-2 | 12.23 | 6.82 | 4.34 | 4.43 | 10.86 | 11.27 | 12.16 | 100 |

**CIFAR-10 (AdvFlow)**

| | Vanilla-1 | Free-1 | Fast-1 | RotNet-1 | Vanilla-2 | Free-2 | Fast-2 | RotNet-2 |
|---|---|---|---|---|---|---|---|---|
| Vanilla-1 | 100 | 5.23 | 3.75 | 3.04 | 7.03 | 6.26 | 5.63 | 5.47 |
| Free-1 | 5.37 | 100 | 18 | 15.05 | 6.76 | 40.53 | 27.55 | 27.68 |
| Fast-1 | 7.07 | 26.44 | 100 | 22.78 | 6 | 32.72 | 32.19 | 30.88 |
| RotNet-1 | 4.52 | 20.31 | 19.39 | 100 | 5.1 | 25.47 | 26.33 | 27.07 |
| Vanilla-2 | 8.82 | 6.45 | 4.66 | 3.96 | 100 | 7.59 | 7.15 | 6.39 |
| Free-2 | 2.96 | 16.4 | 8.84 | 6.82 | 3.68 | 100 | 17.58 | 20.96 |
| Fast-2 | 3.27 | 14.32 | 9.96 | 8.74 | 3.72 | 24.5 | 100 | 26.3 |
| RotNet-2 | 2.55 | 10.58 | 7.32 | 8 | 2.56 | 21.63 | 18.4 | 100 |

**SVHN ($\mathcal{N}$Attack)**

| | Vanilla-1 | Free-1 | Fast-1 | RotNet-1 | Vanilla-2 | Free-2 | Fast-2 | RotNet-2 |
|---|---|---|---|---|---|---|---|---|
| Vanilla-1 | 100 | 2.19 | 1.18 | 1.57 | 9.23 | 1.86 | 0.84 | 1.38 |
| Free-1 | 24.48 | 100 | 8.34 | 13.37 | 24.77 | 23.34 | 7.61 | 12.58 |
| Fast-1 | 28.06 | 24.51 | 100 | 40.31 | 27.57 | 33.16 | 31.51 | 41.08 |
| RotNet-1 | 11.88 | 12.3 | 9.14 | 100 | 10.78 | 15.04 | 9.39 | 21.84 |
| Vanilla-2 | 13.87 | 2.6 | 1.41 | 2.02 | 100 | 2.33 | 1.07 | 1.86 |
| Free-2 | 22.57 | 26.04 | 10.27 | 18.28 | 22.52 | 100 | 10.72 | 18.49 |
| Fast-2 | 23.29 | 22.25 | 24.71 | 33.13 | 25.01 | 30.46 | 100 | 35.06 |
| RotNet-2 | 14.64 | 17.11 | 12.35 | 29.17 | 13.45 | 20.68 | 12.83 | 100 |

**SVHN (AdvFlow)**

| | Vanilla-1 | Free-1 | Fast-1 | RotNet-1 | Vanilla-2 | Free-2 | Fast-2 | RotNet-2 |
|---|---|---|---|---|---|---|---|---|
| Vanilla-1 | 100 | 11.55 | 8.32 | 9.02 | 16 | 10.61 | 8.7 | 8.92 |
| Free-1 | 13.53 | 100 | 15.1 | 20.78 | 14.54 | 31.97 | 18.45 | 21.7 |
| Fast-1 | 16.97 | 37.84 | 100 | 45.92 | 17.99 | 40.12 | 49.84 | 48.75 |
| RotNet-1 | 7.66 | 25.65 | 17.43 | 100 | 8.15 | 23.13 | 23.44 | 33.4 |
| Vanilla-2 | 20.27 | 14.39 | 9.41 | 10.08 | 100 | 11.88 | 10.24 | 10.44 |
| Free-2 | 14.91 | 47.1 | 22.26 | 28.36 | 14.47 | 100 | 28.26 | 31.49 |
| Fast-2 | 12.28 | 31.12 | 28.19 | 37.45 | 13.84 | 32.76 | 100 | 38.98 |
| RotNet-2 | 8.75 | 29.69 | 20.23 | 36.86 | 9.09 | 28.52 | 28.64 | 100 |

Figure 4: Confusion matrix of transferability for adversarial attacks generated by AdvFlow (ours) and $\mathcal{N}$Attack [33]. Each entry shows the success rate of adversarial examples originally generated for the row-wise classifier to attack the column-wise model. Also, the numbers 1 and 2 in the name of each classifier indicates whether it has a Wide-ResNet-32 [63] or ResNet-50 [18] architecture, respectively.

AdvFlow    Clean    $\mathcal{N}$ATTACK                    AdvFlow    Clean    $\mathcal{N}$ATTACK

CIFAR-10 [30]                                              SVHN [40]

Figure 5: Magnified difference and adversarial examples generated by AdvFlow (ours) and $\mathcal{N}$ATTACK [33] alongside the clean data. As can be seen, the adversaries generated by AdvFlow are better disguised in the data, while $\mathcal{N}$ATTACK [33] look noisy (better viewed in digital format).

## C.5   Training Data and Its Effects

In the closing paragraph of Section 3.1 we discussed that while we are using the same training data as the classifier for our flow-based models, it does not have any effect on the performance of generated adversarial examples. This claim is valid since we did not explicitly use this data to either generate adversarial examples or learn their features by which the classifier intends to distinguish them. To support our claim, we replicate the experiments of Table 2 for the case of CIFAR-10 [30]. This time, however, we select 9000 of the test data and pre-train our normalizing flow on them. Then, we evaluate the performance of AdvFlow on the remaining 1000 test data. We then report the attack success rate and the average number of queries in Table 12 for this new scenario (Scenario 2) vs. the original case (Scenario 1).

As can be seen, the performance does not change in general. The little differences between the two cases come from the fact that in Scenario 1, we had 50000 training data to train our flow-based model, while in Scenario 2 we only trained our model on 9000 training data. Also, here we are evaluating AdvFlow performance on 1000 test data in contrast to the whole 10000 test images used in assessing Scenario 1. Finally, it should be noted that we get even a more balanced relative performance for the fair case where we split the training data 50/50 between classifier and flow-based model. However, since the performance of the classifier drops in this case, we only report the unfair situation here.

Table 12: Attack success rate and average (median) of the number of queries to generate an adversarial example for CIFAR-10 [30]. Scenario 1 corresponds to the case where we use the whole CIFAR-10 training data to train our normalizing flow. Scenario 2 indicates the experiment in which we train our flow-based model on 9000 images from CIFAR-10 test data. The architecture of the classifier in all of the cases is Wide-ResNet-32 [63]. Also, all attacks are with respect to $\ell_\infty$ norm with $\epsilon_{\max} = 8/255$.

| Data | Defense | Success Rate(%) ↑ | | Avg. (Med.) of Queries ↓ | |
|---|---|---|---|---|---|
| | | Scenario 1 | Scenario 2 | Scenario 1 | Scenario 2 |
| CIFAR-10 | Vanilla | 99.42 | 98.91 | 950.07 (400) | 949.78 (400) |
| | FreeAdv | 41.21 | 40.22 | 923.58 (200) | 962.31 (200) |
| | FastAdv | 40.22 | 40.93 | 963.77 (200) | 1114.68 (200) |
| | RotNetAdv | 40.67 | 40.32 | 880.86 (200) | 876.57 (400) |

More interestingly, we observe that in case we train our flow-based model on some similar dataset to the original one, we can still get an acceptable relative performance. More specifically, we train our flow-based model on CIFAR-100 [30] dataset instead of CIFAR-10 [30]. Then, we perform AdvFlow on the test data of CIFAR-10 [30]. We know that despite being visually similar, these two datasets have their differences in terms of classes and samples per class.

Table 13 shows the performance of AdvFlow in this case where it is pre-trained on CIFAR-100 instead of CIFAR-10. As the results indicate, we can achieve a competitive performance despite our model being trained on a slightly different dataset. Furthermore, a few adversarial examples from this model are shown in Figure 6. We see that the perturbations are still more or less taking the shape of the data.

Table 13: Attack success rate and average (median) of the number of queries to generate an adversarial example for CIFAR-10 [30] test data. The *train data* row shows the data that is used for training the normalizing flow part of AdvFlow. The architecture of the classifier in all of the cases is Wide-ResNet-32 [63]. Also, all attacks are with respect to $\ell_\infty$ norm with $\epsilon_{\max} = 8/255$.

| Test | Def/Train Data | Success Rate(%) ↑ | | Avg. (Med.) of Queries ↓ | |
|---|---|---|---|---|---|
| | | CIFAR-10 | CIFAR-100 | CIFAR-10 | CIFAR-100 |
| CIFAR-10 | Vanilla | 99.42 | 98.72 | 950.07 (400) | 1198.03 (600) |
| | FreeAdv | 41.21 | 39.95 | 923.58 (200) | 955.05 (200) |
| | FastAdv | 40.22 | 38.83 | 963.77 (200) | 1017.66 (200) |
| | RotNetAdv | 40.67 | 39.28 | 880.86 (200) | 910.55 (200) |

Figure 6: Magnified difference and adversarial examples generated by AdvFlow alongside the clean data using flow-based models trained on CIFAR-100 [30] (left) and CIFAR-10 [30] (right).

CIFAR-100    Clean    CIFAR-10

# D AdvFlow and Its Variations

## D.1 AdvFlow

Algorithm 1 summarizes the main adversarial attack approach introduced in this paper.

---

**Algorithm 1** AdvFlow for inconspicuous black-box adversarial attacks

---

**Input**: Clean data $\mathbf{x}$, true label $y$, pre-trained flow-based model $\mathbf{f}(\cdot)$.
**Output**: Adversarial example $\mathbf{x}'$.
**Parameters**: noise variance $\sigma^2$, learning rate $\alpha$, population size $n_p$, maximum number of queries $Q$.

  1: Initialize $\boldsymbol{\mu}$ randomly.
  2: Compute $\mathbf{z}_{clean} = \mathbf{f}^{-1}(\mathbf{x})$.
  3: **for** $q = 1, 2, \ldots, \lfloor Q/n_p \rfloor$ **do**
  4:      Draw $n_p$ samples from $\boldsymbol{\delta}_z = \boldsymbol{\mu} + \sigma\boldsymbol{\epsilon}$ where $\boldsymbol{\epsilon} \sim \mathcal{N}(\boldsymbol{\epsilon}|\mathbf{0}, I)$.
  5:      Set $\mathbf{z}_k = \mathbf{z}_{clean} + \boldsymbol{\delta}_{zk}$ for all $k = 1, \ldots, n_p$.
  6:      Calculate $\mathcal{L}_k = \mathcal{L}\big(\mathrm{proj}_{\mathcal{S}}\big(\mathbf{f}(\mathbf{z}_k)\big)\big)$ for all $k = 1, \ldots, n_p$.
  7:      Normalize $\hat{\mathcal{L}}_k = \big(\mathcal{L}_k - \mathrm{mean}(\boldsymbol{\mathcal{L}})\big)/\mathrm{std}(\boldsymbol{\mathcal{L}})$.
  8:      Compute $\nabla_{\boldsymbol{\mu}} J(\boldsymbol{\mu}, \sigma) = \frac{1}{n_p \cdot \sigma} \sum_{k=1}^{n_p} \hat{\mathcal{L}}_k \boldsymbol{\epsilon}_k$.
  9:      Update $\boldsymbol{\mu} \leftarrow \boldsymbol{\mu} - \alpha\nabla_{\boldsymbol{\mu}} J(\boldsymbol{\mu}, \sigma)$.
10: **end for**
11: Output $\mathbf{x}' = \mathrm{proj}_{\mathcal{S}}\big(\mathbf{f}(\mathbf{z}_{clean} + \boldsymbol{\mu})\big)$.

---

## D.2 Greedy AdvFlow [9]

We can modify Algorithm 1 so that it stops upon reaching a data point where it is adversarial. To this end, we only have to actively check whether we have generated a data sample for which the C&W cost of Eq. (1) is zero or not. Besides, instead of using all of the generated samples to update the mean of the latent Gaussian $\boldsymbol{\delta}_z$, we can select the top-$K$ for which the C&W loss is the lowest. Then, we update the mean by taking the average of these latent space data points. Applying these changes, we get a new algorithm coined *Greedy AdvFlow*. This approach is given in Algorithm 2.

---

**Algorithm 2** Greedy AdvFlow for inconspicuous black-box adversarial attacks

---

**Input**: Clean data $\mathbf{x}$, true label $y$, pre-trained flow-based model $\mathbf{f}(\cdot)$.
**Output**: Adversarial example $\mathbf{x}'$.
**Parameters**: noise variance $\sigma^2$, voting population $K$, population size $n_p$, maximum number of queries $Q$.

  1: Initialize $\boldsymbol{\mu}$ randomly.
  2: Compute $\mathbf{z}_{clean} = \mathbf{f}^{-1}(\mathbf{x})$.
  3: **for** $q = 1, 2, \ldots, \lfloor Q/n_p \rfloor$ **do**
  4:      Draw $n_p$ samples from $\boldsymbol{\delta}_z = \boldsymbol{\mu} + \sigma\boldsymbol{\epsilon}$ where $\boldsymbol{\epsilon} \sim \mathcal{N}(\boldsymbol{\epsilon}|\mathbf{0}, I)$.
  5:      Set $\mathbf{z}_k = \mathbf{z}_{clean} + \boldsymbol{\delta}_{zk}$ for all $k = 1, \ldots, n_p$.
  6:      Calculate $\mathcal{L}_k = \mathcal{L}\big(\mathrm{proj}_{\mathcal{S}}\big(\mathbf{f}(\mathbf{z}_k)\big)\big)$ for all $k = 1, \ldots, n_p$.
  7:      **if** any $\mathcal{L}_k$ becomes 0 **then**:
  8:          Output the $\mathbf{x}' = \mathrm{proj}_{\mathcal{S}}\big(\mathbf{f}(\mathbf{z}_k)\big)$ for which $\mathcal{L}_k = 0$ as the adversarial example.
  9:          break
10:      **end if**
11:      Find the top-$K$ samples $\mathbf{z}_k$ with the lowest score $\mathcal{L}_k$.
12:      Update $\boldsymbol{\mu} \leftarrow \frac{1}{K} \sum_{k \in \mathrm{top}-K} \boldsymbol{\delta}_{zk}$.
13: **end for**

---

Table 14 shows the performance of the proposed method with respect to AdvFlow. As can be seen, this way we can improve the success rate and number of required queries.

Table 14: Attack success rate and average (median) of the number of queries to generate an adversarial example for CIFAR-10 [30], and SVHN [40]. The architecture of the classifier in all of the cases is Wide-ResNet-32 [63]. Also, all attacks are with respect to $\ell_\infty$ norm with $\epsilon_{\max} = 8/255$. The hyperparameters used for Greedy AdvFlow are the same as Table 10. In each iteration, we select the top-4 data samples to update the mean.

| Data | Defense | Success Rate(%) ↑ | | Avg. (Med.) of Queries ↓ | |
|---|---|---|---|---|---|
| | | Greedy AdvFlow | AdvFlow | Greedy AdvFlow | AdvFlow |
| CIFAR-10 | Vanilla | 99.12 | 99.42 | 991.98 (460) | 950.07 (400) |
| | FreeAdv | 41.06 | 41.21 | 842.37 (180) | 923.58 (200) |
| | FastAdv | 40.06 | 40.22 | 904.78 (200) | 963.77 (200) |
| | RotNetAdv | 40.50 | 40.67 | 821.80 (180) | 880.86 (200) |
| SVHN | Vanilla | 92.40 | 90.42 | 1305.06 (540) | 1582.87 (800) |
| | FreeAdv | 52.57 | 50.63 | 816.54 (200) | 1095.68 (200) |
| | FastAdv | 43.03 | 41.39 | 781.06 (240) | 1046.45 (400) |
| | RotNetAdv | 45.43 | 44.37 | 653.82 (160) | 923.59 (200) |

## D.3 AdvFlow for High-resolution Images

Despite their ease-of-use in generating low-resolution images, high-resolution image generation with normalizing flows is computationally demanding. This issue is even more pronounced in the case of images with high variabilities, such as the ImageNet [46] dataset, which may require a lot of invertible transformations to model them. To cope with this problem, we propose an adjustment to our AdvFlow algorithm. Instead of generating the image in the high-dimensional space, we first map it to a low-dimension space using bilinear interpolation. Then, we perform the AdvFlow algorithm to generate the set of candidate examples. Next, we compute the adversarial perturbations in the low-dimensional space and map them back to their high-dimensional representation using bilinear upsampling. These perturbations are then added to the original target image, and the rest of the algorithm continues as before. Figure 7 shows the block-diagram of the proposed solution for high-resolution data. Moreover, the updated AdvFlow procedure is summarized in Algorithm 3. Changes are highlighted in red.

To test the performance of the proposed approach, we pre-train a flow-based model on a $64 \times 64$ version of ImageNet [46]. The normalizing flow architecture and training hyperparameters are as shown in Table 4. Furthermore, we use bandits with data-dependent priors [23] and $\mathcal{N}$ATTACK [33] to compare our model against them. For bandits, we use the tuned hyperparameters for ImageNet [46] in the original paper [23]. Also, for $\mathcal{N}$ATTACK [33] and AdvFlow we observe that the hyperparameters in Tables 8 and 10 work best for the vanilla architectures in this dataset. Thus, we kept them as before.

Figure 7: AdvFlow adjustment for high-resolution images. Instead of working with high-dimensional image, we downsample them. Then after generating candidate low-resolution perturbations, we map them to high-dimensions using a bilinear upsampler.

**Algorithm 3** AdvFlow for high-resolution black-box attack

---

**Input**: Clean data $\mathbf{x}$, true label $y$, pre-trained flow-based model $\mathbf{f}(\cdot)$.
**Output**: Adversarial example $\mathbf{x}'$.
**Parameters**: noise variance $\sigma^2$, learning rate $\alpha$, population size $n_p$, maximum number of queries $Q$.

  1: Initialize $\boldsymbol{\mu}$ randomly.
  2: Downsample $\mathbf{x}$ and save it as $\mathbf{x}_{\text{low}}$.
  3: Compute $\mathbf{z}_{clean} = \mathbf{f}^{-1}(\mathbf{x}_{\text{low}})$.
  4: **for** $q = 1, 2, \ldots, \lfloor Q/n_p \rfloor$ **do**
  5:     Draw $n_p$ samples from $\boldsymbol{\delta}_z = \boldsymbol{\mu} + \sigma\boldsymbol{\epsilon}$ where $\boldsymbol{\epsilon} \sim \mathcal{N}(\boldsymbol{\epsilon}|\mathbf{0}, I)$.
  6:     Set $\mathbf{z}_k = \mathbf{z}_{clean} + \boldsymbol{\delta}_{zk}$ for all $k = 1, \ldots, n_p$.
  7:     Compute and upsample $\boldsymbol{\gamma}_k = \mathbf{f}(\mathbf{z}_k) - \mathbf{x}_{\text{low}}$ for all $k = 1, \ldots, n_p$.
  8:     Calculate $\mathcal{L}_k = \mathcal{L}\big(\text{proj}_{\mathcal{S}}(\boldsymbol{\gamma}_k + \mathbf{x})\big)$ for all $k = 1, \ldots, n_p$.
  9:     Normalize $\hat{\mathcal{L}}_k = \big(\mathcal{L}_k - \text{mean}(\boldsymbol{\mathcal{L}})\big)/\text{std}(\boldsymbol{\mathcal{L}})$.
10:     Compute $\nabla_{\boldsymbol{\mu}} J(\boldsymbol{\mu}, \sigma) = \frac{1}{n_p \cdot \sigma} \sum_{k=1}^{n_p} \hat{\mathcal{L}}_k \boldsymbol{\epsilon}_k$.
11:     Update $\boldsymbol{\mu} \leftarrow \boldsymbol{\mu} - \alpha\nabla_{\boldsymbol{\mu}} J(\boldsymbol{\mu}, \sigma)$.
12: **end for**
13: Upsample $\boldsymbol{\gamma} = \mathbf{f}(\mathbf{z}_{clean} + \boldsymbol{\mu}) - \mathbf{x}_{\text{low}}$.
14: Output $\mathbf{x}' = \text{proj}_{\mathcal{S}}(\boldsymbol{\gamma} + \mathbf{x})$.

---

We use the nominated black-box methods to attack a classifier in less than $10,000$ queries. We use pre-trained Inception-v3 [52], ResNet-50 [18], and VGG-16 [50] classifiers available on `torchvision` as our vanilla target models. Also, a defended ResNet-50 [18] model trained by fast adversarial training [60] with FGSM ($\epsilon = 4/255$) is used for evaluation. This model is available online on the official repository of fast adversarial training.[11]

Table 15 shows our experimental results on the ImageNet [46] dataset. As can be seen, we get similar results to CIFAR-10 [30] and SVHN [40] experiments in Table 2: in case of vanilla architectures, we are performing slightly worse than $\mathcal{N}$ATTACK [33], while in the defended case we improve their performance considerably. It should also be noted again that in all of the cases we are generating adversaries that look like the original data, and come from the same distribution. This property is desirable in confronting adversarial example detectors. Figure 8 depicts a few adversarial examples generated by AdvFlow compared to $\mathcal{N}$ATTACK [33] for a vanilla Inception-v3 [52] DNN classifier. As seen, the AdvFlow perturbations tend to take the shape of the data to reduce the possibility of changing the underlying data distribution. In contrast, $\mathcal{N}$ATTACK perturbations are pixel-level, independent additive noise that cause the adversarial example distribution to become different from that of the data.

Table 15: Attack success rate and average (median) of the number of queries needed to generate an adversarial example for ImageNet [46]. For a fair comparison, we first find the samples where all the attack methods are successful, and then compute the average (median) of queries for these samples. Note that for $\mathcal{N}$ATTACK and AdvFlow we check whether we arrived at an adversarial point every 200 queries, hence the medians are multiples of 200. All attacks are with respect to $\ell_\infty$ norm with $\epsilon_{\max} = 8/255$. * The accuracy is computed with respect to the 1000 test data used for attack evaluation.

| Data | Arch. | Attack Acc*(%) | Bandits [23] / $\mathcal{N}$ATTACK [33] / SimBA [16] / AdvFlow (ours) | |
|---|---|---|---|---|
| | | | Success Rate(%) ↑ | Avg. (Med.) of Queries ↓ |
| ImageNet | Inception-v3 | 99.20 | 87.80 / **95.06** / 80.95 / 87.50 | 1034.41 (430) / **680.52 (400)** / 1481.12 (1142) / 1516.34 (800) |
| | VGG16 | 92.50 | 95.46 / **99.57** / 97.95 / 97.51 | 541.64 (**166**) / **395.61** (200) / 608.42 (486) / 1239.03 (600) |
| | Van. ResNet | 95.00 | 95.79 / **99.47** / 98.42 / 95.58 | 948.90 (**364**) / **604.31** (400) / 701.92 (494) / 1501.13 (800) |
| | Def. ResNet | 71.50 | 50.77 / 33.99 / 47.55 / **57.20** | 914.58 (404) / 2170.82 (1200) / 969.91 (696) / **381.97 (200)** |

AdvFlow (ours)          Clean Image          $\mathcal{N}$ATTACK

Figure 8: Magnified difference and adversarial examples generated by AdvFlow (ours) and $\mathcal{N}$ATTACK [33] alongside the clean data for ImageNet [46] dataset. The target network architecture is Inception-v3 [52].

### D.4 AdvFlow for People in a Hurry!

Alternatively, one can use the plain structure of AdvFlows for black-box adversarial attacks. To this end, we are only required to initialize the normalizing flow randomly. This way, however, we will be getting random-like perturbations as in $\mathcal{N}$ATTACK [33] since we have not trained the flow-based model. Using this approach, we can surpass the performance of the baselines in vanilla DNNs. In fact, giving away a little bit of performance is the price we pay to force the perturbation to have a data-like structure so that the adversaries have a similar distribution to the data. Table 16 summarizes the performance of randomly initialized AdvFlow in contrast to bandits with data-dependent priors [23] and $\mathcal{N}$ATTACK [33] for vanilla ImageNet [46] classifiers. Also, Figure 9 shows a few adversarial examples generated by randomly initialized AdvFlows in this case.

Table 16: Attack success rate and average (median) of the number of queries needed to generate an adversarial example for ImageNet [46]. For a fair comparison, we first find the samples where all the attack methods are successful, and then compute the average (median) of queries for these samples. Note that for $\mathcal{N}$ATTACK and AdvFlow we check whether we arrived at an adversarial point every 200 queries, hence the medians are multiples of 200. All attacks are with respect to $\ell_\infty$ norm with $\epsilon_{\max} = 8/255$. * The accuracy is computed with respect to the 1000 test data used for attack evaluation.

| Arch. | Attack Acc*(%) | Bandits [23] / $\mathcal{N}$ATTACK [33] / SimBA [16] / AdvFlow (Random Init.) | |
|---|---|---|---|
| | | Success Rate(%) ↑ | Avg. (Med.) of Queries ↓ |
| Inception-v3 | 99.20 | 87.80 / 95.06 / 80.95 / **97.78** | 1081.13 (452) / 745.10 (400) / 1537.71 (1173) / **375.00** (**200**) |
| VGG16 | 92.50 | 95.46 / **99.57** / 97.95 / 99.35 | 586.14 (174) / 418.33 (**200**) / 637.96 (503) / **299.89** (**200**) |
| Van. ResNet | 95.00 | 95.79 / **99.47** / 99.16 / 98.42 | 1053.44 (415) / 672.53 (400) / 758.32 (523) / **370.63** (**200**) |

Pre-trained AdvFlow      Clean Image      Random AdvFlow

Figure 9: Magnified difference and adversarial examples generated by AdvFlow in trained and un-trained scenarios for ImageNet [46] dataset. The target network architecture is Inception-v3 [52].

## Footnotes

[5]github.com/VLL-HD/FrEIA

[6]github.com/pokaxpoka/deep_Mahalanobis_detector

[7]github.com/EvZissel/Residual-Flow

[8]github.com/mahyarnajibi/FreeAdversarialTraining

[9]github.com/anonymous-sushi-armadillo

[10]github.com/hendrycks/ss-ood

[11]github.com/anonymous-sushi-armadillo