[Reviews · NeurIPS 2020]

Review 1

Summary and Contributions: This paper proposes a new score-based black-box attack method. A flow-based generative model is used to model the probability distribution of data. The adversarial examples are searched over the latent space of the flow-based model, making them hard to detect. Experimental results on CIFAR-10 and SVHN demonstrate the effectiveness of the proposed method over two baselines.

Strengths: This paper introduces the idea of using flow-based generative models for effective black-box adversarial attacks. The method is technically sound and empirically effective.

Weaknesses: Although the idea of using a flow-based generative model for adversarial attacks is new, there already exist other works on using other types of generative models for adversarial attacks. The authors have reviewed related works in Sec. 2, and the main claim is that the proposed model can be pre-trained and adapted to any classifier. Why do other generative models (e.g., GANs) cannot be used in this way? Another concern is about the experiments of this paper. Only two baselines (Bandits and Nattack) are adopted for comparison. There are many recent works in this field. The authors are encouraged to compare with SOTA to show the effectiveness.

Correctness: The proposed method is technically correct. The experimental methodology is also correct, though it could be improved (e.g., more experiments on ImageNet are encouraged to include).

Clarity: Yes.

Relation to Prior Work: Yes

Reproducibility: Yes

Additional Feedback: Comments after rebuttal: Thanks for the efforts when preparing the rebuttal. Now it's much clearer why normalizing flows are better for black-box adversarial attacks than other generative models. The additional results on the comparison with SimBA are also promising. Although the idea of stacking a NF model before the classifier is simple and well-motivated, I found that the adversarial attack method is almost the same as Nattack (in Eq.(12)). Then the main technical contributions lie in using the NF model, which could be insufficient since a previous work also tried to use an Auto-encoder [48] before the classifier (the authors have discussed the differences between [48], but the the idea is similar). My further suggestion on this paper is to conduct experiments on large-scale datasets (e.g., ImageNet), which could better show the effectiveness of the proposed method.


Review 2

Summary and Contributions: This work generates adversarial samples that follow the structure of the original data distribution. They are the first to estimate this data distribution with a normalizing flow and then perturb the flow's latent distribution to make it represent a modified data distribution, the adversarial distribution. The induced structure in the adversarial distribution is somewhat less likely to be detected by typical defendant classifiers than previous approaches approximating the adversarial distribution. The approach to create adversarials using generative models is a promising direction that allows arguments about the structure of the samples. This method provides better adversarials than previous SOTA for defended classifiers, and performs comparably for vanilla classifiers.

Strengths: The adversarial distribution learnt by this method can represent coordinated changes across an image instead of changing pixels independently. This allows adversarials to lie closer to the original data, smeering the line between data and adversarial distribution.

Weaknesses: The performance on vanilla and defended classifiers is different: AdvFlow only outperforms for the defended case. It is unclear to me how an affine transformation of the latent space affects the distribution represented by a normalizing flow. IMO, this is out of the scope of this paper and an open problem for NFs.

Correctness: Lemma 3.1 certainly holds – but Corollary 3.1.1 does not follow without further analysis: Is the learnt delta_z in the valid limit of the Taylor expansion?

Clarity: Yes, the paper is self-contained and guides the reader well. I would like to hear the answer to one technical question: Why can sigma not be learnt jointly with mu? --> Update: Thanks for the answer that it is a hyperparameter that might be added to the list of variables.

Relation to Prior Work: Yes, this work is the first to directly learn the adversarial distribution by slightly modifying a learnt representation of the data distribution using a flow model. The empirical findings mainly compare with the approach to explicitly represent the adversarial distributino. It would be interesting to compare to state of the art adversarial samples on the same architecture. Since the general idea is very promising, I don't deem it necessary that this line of work produces SOTA in its first steps. --> Update: The comparison with recent methods indicates that the proposed method is currently the most successful against defended classifiers, less against vanilla classifiers.

Reproducibility: Yes

Additional Feedback: I sincerely like the idea to systematically shift the data distribution in the latent space to produce a meaningful adversarial distribution. The results convince me that this learnt distribution is close to the data. Large-scale experiments such as ImageNet are computationally very expensive using NFs. I don't deem it necessary to perform such experiments to publish a neat idea, which is successful on other simpler yet non-trivial datasets.


Review 3

Summary and Contributions: The paper introduces AdvFlow, the first black-box adversarial attack that leverages normalizing flows in modeling the adversarial data distribution. A lemma is proved showing that perturbations with dependent elements can be generated (unlike some pre-existing black-box attack methods). It is shown that using a normalizing flow model, adversarial examples that have a distribution similar to that of the data can be generated. This allows the black-box attack to mislead adversarial example detectors better than pre-existing work since often detectors assume adversarial examples come from a distribution dissimilar to that of clean data.

Strengths: The theoretical set-up is sound, very closely mirroring NAttack [27]. Table 1 demonstrates some promising evidence that by modeling adversarial examples as coming from a distribution that is very similar to that of the input data, an adversarial detector that assumes adversarial examples come from a significantly different distribution can be fooled. Table 2 shows promising empirical evidence that AdvFlow is more query efficient and has a higher attack success rate than several competing methods on adversarially trained models.

Weaknesses: I believe the evaluation is not very convincing as it is right now. Namely, a particularly strong (very query-efficient, high success rate) black-box method is missing: SimBA [a]. Additionally, although Table 1 is promising, I would like to see a stronger evaluation or stronger evidence of some kind that the detector is being fooled by the distributional properties of the adversarial examples. At the very least, I think Table 1 should test against several detectors and against several black-box attacks (as opposed to one detector and one baseline attack). In general, as of right now the paper is on the level of a promising novelty. It is interesting that one can generate adversarial examples with a distribution similar to that of inputs, but evaluation for its usefulness in fooling detectors is unconvincing. [a] Simple Black-box Adversarial Attacks (https://arxiv.org/pdf/1905.07121.pdf)

Correctness: The empirical methodology appears to be correct and the tables and figures are reasonable; the claims made are correct.

Clarity: The paper is fairly well-written; explanations are clear.

Relation to Prior Work: Yes, related work is discussed to the extent that it should be and new contributions are mentioned explicitly at the end of Section 1.

Reproducibility: Yes

Additional Feedback: I believe the paper as it is lacks in proper evaluation against detectors: it would be interesting to see it fool multiple (not just one) detector against a myriad of black-box attacks such as SimBA [a] (mentioned above). [Post-rebuttal Update] I have reviewed the authors' rebuttal, and appreciate their comparison against the strong blackbox attack SimBA and the inclusion of evaluation against multiple detectors. Based on these results, it appears that the attack may indeed be the new SoTA. I strongly encourage the authors to include said results in the updated version of their paper, and potentially make them even more extensive (e.g. running on SVHN, to get a complete version of the original Table 1). Given that some of my original concerns about evaluation have been assuaged, I have updated my rating accordingly.


Review 4

Summary and Contributions: The authors proposed a novel black-box adversarial attack method. They exploit normalizing flows on some data distributions to make their adversarial examples more robust. They build their algorithm upon the natural evolution strategies (NES) and experiment on SVHN and CIFAR-10.

Strengths: - The problem they are solving (black-box adversarial attack) is a standard problem that has been studied extensively. This work provides another insight on how we should design our attack space and may help us better understand the vulnerability of the deep networks. - The idea of using normalizing flows to add reasonable perturbation is novel and interesting. - They show that their attack is stealthier against defended networks, when compared to Bandits and N attack.

Weaknesses: - The maths are not easy to understand for an average computer vision researcher. This reviewer has difficulty understanding the role of AdvFlow, NES and N attack, thus has difficulty assessing the novelty contribution of this paper. - Figure 1 is attacking VGG19 on CelebA but there is no corresponding experiment in the paper or the supplementary material. - Discussions on possible defense methods? - I am not familiar with prior black-box attacking methods, but comparing to N-attack and bandits seems limited. If the method is built upon N attack and bandits, then it could be natural to expect that it outperforms N attack and bandits.

Correctness: The author did not check maths. However, the claims and the experiment results are consistent.

Clarity: All the parts except methodology are easy to parse and comprehend. The technical details of the method may need further work if we want to convey the methodology to a broad range of readers.

Relation to Prior Work: The authors mainly talked about N attack [27] and bandits [17]. I guess this is because their method could be a direct successor of these two methods. However, it is not clear whether there are other black-box attack methods that take a much different approach and how they compare to this work in terms of performance.

Reproducibility: Yes

Additional Feedback: After reading the author feedback and other reviewers comments I decided not to change my score.

[Author Response · NeurIPS 2020]

First, we thank all the reviewers for their invaluable assessment of our paper in this challenging time. As they agree, the general idea of AdvFlows is sound and promising, and the paper is well-written and self-contained. In the following, we address some of the questions raised by the reviewers as much as time and space allows.

**Overview** The final goal of designing adversarial attacks is gaining a better insight into the pitfalls of DNNs, ultimately alleviating such threats. In this regard, designing attacks with a statistical flavor is extremely valuable as they: 1) provide a unifying framework of modeling DNNs' adversarial vulnerability, and more importantly, 2) help in establishing the required connection with mature fields like high-dimensional statistics to use their results in finding the ultimate solution to making DNNs more robust. Having these in mind, we have come up with AdvFlow that can be viewed as an important step in this direction.

**Why NFs and not GANs?** The ability of *Normalizing flows* (NF) for efficient inference and sampling, as well as their straightforward and stable training, made them an ideal candidate for our purpose of designing a black-box attack with a statistical perspective. Note that *generative adversarial networks* (GAN) have many disadvantages for use in the current framework: 1) It is known that GANs suffer from *mode collapse*, where they fail to represent different modes of data equally well. In contrast, flow-based models are trained to maximize the log-likelihood, and as such, they cover different modes of data better. 2) Finding the latent space representation of data in GANs requires solving a non-convex optimization problem by back-propagating through the model for every new attack. However, the proposed NF models are invertible by design, and to find the latent space representation of an image, one only needs to query the model. 3) More importantly, GANs neither represent an explicit distribution nor enable inference and density computation. The current design, however, enables further investigation of the attacker distribution properties in the future.

**Attack strength** The primary purpose of the current work is to convey the idea of blending statistical methods like normalizing flows and adversarial attacks so that we can better understand such threats. Thus, we aimed to compare with recent, but widely recognized black-box attacks for comparison. Nevertheless, by doing more rigorous hyper-parameter tuning or adding extra variables (like $\sigma$ as correctly indicated by R2), the results can be improved further.[1]

**Ablation study on adversarial example detectors** To provide more reliable evidence that AdvFlow's distributional properties are fooling the adversarial example detectors, we perform the following ablation study. First, we use an untrained (denoted by un.) AdvFlow model that is initialized randomly. Then, we use the trained version (denoted by tr.) of the same architecture to perform black-box attacks. Using examples generated by these two models, we then train adversarial example detectors to spot the adversaries from clean images. In the paper, we used the Mahalanobis detector [26], a well-known SOTA adversarial example detector. For the sake of completeness, we also add LID [31] (the previous SOTA) and Res-Flow [58] (the recently introduced SOTA) alongside Mahalanobis detector. We compare our results with $\mathcal{N}$ATTACK, *which also approaches the black-box adversarial attack from a distributional perspective* for a fair comparison. The results are given in Table 1. As shown, only if we pre-train our method on clean data, we can fool the detectors. This is indicating that the attacker's distributional properties are fooling the detectors.

**Performance comparison with SimBA [59]** Note that SimBA [59] was not included in the original manuscript as it is designed for efficient $\ell_2$ attacks. At the time of writing the paper, it was not clear how it can be generalized to $\ell_\infty$. Not until after the NeurIPS deadline did the authors include a generalized version for $\ell_\infty$, alongside the explanations.[2] We repeat the CIFAR-10 experiments of the paper using the recent version of SimBA-DCT, and report the results in Table 2. For a fair comparison, we compute the average and median of queries on examples where both methods have succeeded. As seen, we get similar results to Table 2 of the paper, outperforming SimBA in defended baselines.[3]

Table 1: Adv. example detection on CIFAR-10.

| Detector | AUROC(%)↑ | | |
|---|---|---|---|
| | $\mathcal{N}$ATTACK | Ours (un.) | Ours (tr.) |
| LID [31] | 78.69 | 84.39 | **57.99** |
| Mah. [26] | 97.95 | 99.50 | **66.85** |
| Res. [58] | 97.90 | 99.40 | **67.03** |

Table 2: Performance comparison with SimBA [59] on CIFAR-10.

| Defense | Success Rate(%)↑ | | Query Avg. ↓ | | Query Med. ↓ | |
|---|---|---|---|---|---|---|
| | SimBA | Ours | SimBA | Ours | SimBA | Ours |
| Vanilla | **99.98** | 99.42 | **238.08** | 949.55 | **126** | 400 |
| FreeAdv | 35.52 | **41.21** | 497.97 | **458.35** | 256 | **200** |
| FastAdv | 35.07 | **40.22** | 469.15 | 477.77 | 245 | **200** |
| RotNetAdv | 35.63 | **40.67** | 499.75 | **453.26** | 267 | **200** |

## Footnotes

[1] Note that some of the current SOTA results in black-box adversarial attacks come from the attacker's knowledge about the gradients of the target classifier using substitute models. However, once the target changes its training procedure (e.g., from vanilla to adversarial training), the performance of such methods drop significantly. In contrast, **our method is trained only on clean data and does not depend on any substitute network**. As such, it has a considerable advantage against these methods that are currently prevalent.

[2] See the official repo. of SimBA, where it clearly is indicated that the $\ell_\infty$ attack is added on 2020/06/22, after NeurIPS deadline.

[3] The results of Table 1 and 2 (as well as SVHN) will be added to the camera-ready version.

[58] Zisselman and Tamar. "Deep Residual Flow for Out-of-Distribution Detection." *CVPR*, 2020.

[59] Guo et al. "Simple Black-box Adversarial Attacks." *ICML*, 2019.


[Meta-Review · NeurIPS 2020]

This paper presents a new score-based black-box attack method. It uses normalizing flow to estimate the adversarial data distribution. The adversarial examples are searched over the latent space of the flow-based model, making them hard to detect. Experimental results on CIFAR-10 and SVHN demonstrate its effectiveness over two baselines. Overall, it makes valuable contributions. The rebuttal addressed most of the concerns, with additional results. In the final version, the authors need to include said results and potentially make them even more extensive (e.g. running on SVHN, to get a complete version of the original Table 1).